# Batch Normalization Increases Adversarial Vulnerability

## Abstract

Batch normalization (BN) is often used in an attempt to stabilize and accelerate training in deep neural networks. In many cases it indeed decreases the number of parameter updates required to achieve low training error. However, it also reduces robustness to small adversarial input perturbations and common corruptions by double-digit percentages, as we show on five standard datasets. Furthermore, we find that substituting weight decay for BN is sufficient to nullify a relationship between adversarial vulnerability and the input dimension. A recent mean-field analysis found that BN induces gradient explosion when used on multiple layers, but this does not fully explain the vulnerability we observe, given that it occurs for a single BN layer. We argue that the main cause is the tilting of the decision boundary with respect to the nearest-centroid classifier along input dimensions of low variance. As a result, the numerical stability constant of BN acts as an important meta-parameter that can be tuned to recover some robustness at the cost of standard test accuracy. We explain this mechanism explicitly for linear models but find that it still holds for nonlinear models.

## 1 Introduction

BN is a standard component of modern deep neural networks, and tends to make the training process less sensitive to the choice of hyperparameters in many cases (Ioffe & Szegedy, 2015). While ease of training is desirable for model developers, an important concern among stakeholders is that of model robustness during deployment to plausible, previously unseen inputs. The adversarial examples phenomenon has exposed unstable predictions across state-of-the-art models (Szegedy et al., 2014). This has led to a variety of methods that aim to improve robustness, but doing so effectively remains a challenge (Athalye et al., 2018; Schott et al., 2019; Hendrycks & Dietterich, 2019; Jacobsen et al., 2019a). We believe that a prerequisite to developing methods that increase robustness is an understanding of factors that reduce it.

Approaches for improving robustness often begin with existing neural network architectures—that use BN—and patch them against specific attacks, e.g., through inclusion of adversarial examples during training (Szegedy et al., 2014; Goodfellow et al., 2015; Kurakin et al., 2017; Madry et al., 2018). An implicit assumption is that BN itself does not reduce robustness, however, recent initialization-time analyses have shown that it causes exploding gradients, and increased sensitivity to input perturbations as the network depth increases (Yang et al., 2019; Labatie, 2019). In this work, we consider the impact of BN in practical scenarios in terms of robustness to *common corruptions* (Hendrycks & Dietterich, 2019) and *adversarial examples* (Szegedy et al., 2014), finding that BN induced sensitivity remains a concern even in cases where its use appears benign on the basis of clean test accuracy, and when only one BN layer is used.

The frequently made observation that adversarial vulnerability can scale with the input dimension (Goodfellow et al., 2015; Gilmer et al., 2018; Simon-Gabriel et al., 2019) highlights the importance of identifying regularizers as more than merely a way to improve test accuracy. In particular, BN was a confounding factor in Simon-Gabriel et al. (2019), making the results of their initialization-time analysis hold after training. By adding $\ell_2$ regularization and removing BN, we show that there is no *inherent* relationship between adversarial vulnerability and the input dimension.

## 2 BATCH NORMALIZATION

We briefly review how BN modifies the hidden layers' pre-activations $h$ of a neural network. We use the notation of Yang et al. (2019), where $\alpha$ is an index for units in a layer $l$, and $i$ for a mini-batch of $B$ samples from the dataset; $N_l$ denotes the number of units in layer $l$, $W^l$ is the matrix of weights and $b^l$ is the vector of biases that parametrize layer $l$. The batch mean is defined as $\mu_\alpha = \frac{1}{B} \sum_i h_{\alpha i}$, and the variance is $\sigma_\alpha^2 = \frac{1}{B} \sum_i (h_{\alpha i} - \mu_\alpha)^2$. In the BN procedure, the mean $\mu_\alpha$ is subtracted from the pre-activation of each unit $h_{\alpha i}^l$ (consistent with Ioffe & Szegedy (2015)), the result is divided by the standard deviation $\sigma_\alpha$ plus a small constant $c$ to prevent division by zero, then scaled and shifted by the learned parameters $\gamma_\alpha$ and $\beta_\alpha$, respectively. This is described in equation 1, where a per-unit nonlinearity $\phi$, e.g., ReLU, is applied after the normalization.

$$h_i^l = W^l \phi(\tilde{h}_i^{l-1}) + b^l, \qquad \tilde{h}_{\alpha i}^l = \gamma_\alpha \frac{h_{\alpha i} - \mu_\alpha}{\sqrt{\sigma_\alpha^2 + c}} + \beta_\alpha. \tag{1}$$

This procedure introduces complications, however. Consider two mini-batches that differ by only a *single* example: due to the induced batch-wise nonlinearity, they will have different representations of *all* examples. These differences are amplified by stacking BN layers, and were shown to cause exploding gradients at initialization (Yang et al., 2019). Conversely, normalization of intermediate representations for two different training inputs impairs the ability to distinguish definite examples that ought to be classified with a large prediction margin (as judged by an "oracle"), from more ambiguous instances. The last layer of a discriminative neural network, in particular, is typically a linear decoding of class label-homogeneous clusters, and thus makes use of information contained in the mean and variance at this stage for classification. In light of these observations, we begin in our analysis by adding a single BN layer to models trained by gradient descent (GD). This is the most favorable scenario according to the analysis of Yang et al. (2019), where more layers and a smaller mini-batch size exacerbate the exploding gradients.

## 3 BOUNDARY TILTING

Tanay & Griffin (2016) relate the adversarial vulnerability of linear classifiers to the tilting angle $\theta$ of the decision boundary w.r.t. the nearest-centroid classifier. Following their setup, we examine how BN affects this angle in a simple linear model, and then show that increasing model complexity cannot "undo" this vulnerability.

Consider the binary classification task of identifying two different types of input $x$ subject to Gaussian noise with a linear classifier $w^\top x + b$. This can be modeled by the class-conditional distribution $p(x|y = j) = \mathcal{N}(\nu^j, \Sigma)$ with label $y \sim \text{Ber}(0.5)$. The Bayes-optimal solution to this problem is given by the weight vector $w = \Sigma^{-1}(\nu^0 - \nu^1)$, and $b = \frac{1}{2}(\nu^1 + \nu^0)^\top \Sigma^{-1}(\nu^1 - \nu^0) + \log \frac{p(y=0)}{p(y=1)}$, where $p(y)$ denotes the marginal probability for the label $y$ (see e.g. (Jordan, 1995)), while the nearest-centroid classifier is defined by $w^* = \nu^0 - \nu^1$.

We analyze the effect of batch-normalizing the input to the classifier for this problem (i.e., $h_{\alpha i} = x_{\alpha i}$), first in the simplest setting where $\gamma_\alpha = 1, \beta_\alpha = 0 \,\forall \alpha$. We select the class distribution means $\nu^j$ to be symmetric around zero, so that the batch mean computed by BN is $\mu_\alpha = 0 \,\forall \alpha$. The batch-normalized linear classifier is thus defined as: $f(x) = \frac{w^\top x + b}{\sqrt{\sigma^2 + c}}$. By construction of our synthetic dataset, the variance of the batch can be deduced from the data

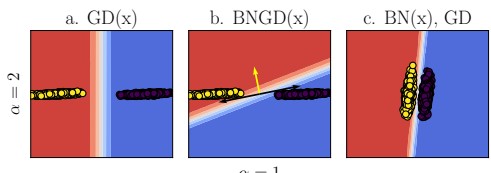

Figure 1: A dataset with one task-relevant ($\alpha = 1$) and one task-irrelevant dimension ($\alpha = 2$). Normalization aligns the decision boundary with the Bayes solution (indicated by arrows in "BNGD"), but this minimizes the averaged distance between the points and the boundary, maximizing adversarial vulnerability. Compared with the decision boundary of a linear model ($\theta \approx 0°$), the batch-normalized model has $\theta = 66.7°$. On the right is the dataset seen by the BNGD classifier. We use $\Sigma_{11} = 1$, $\Sigma_{22} = 0.01$, $\Sigma_{12} = \Sigma_{21} = 0.05$, $\nu^0 = [-5, 0]$, and $\nu^1 = [5, 0]$.

Table 1: As predicted by the theory, batch-normalized gradient descent (BNGD) yields a tilted decision boundary w.r.t. the nearest-centroid classifier, regardless of the affine parameters being learned or fixed. We report the tilting angle ($\theta$) and accuracies of linear models trained on MNIST 3 vs. 7 for vanilla GD, GD with L2 weight decay "WD"($\lambda = 0.1$), and BNGD. Affine = "F" indicates $\gamma = 1$ and $\beta = 0$, whereas "T" means they are randomly initialized and learnable. AWGN = $\mathcal{N}(0, 1)$, FGSM used with $\epsilon = 1/10$. Entries are the mean and its standard error over five random seeds.

| Model | Test Acc. | AWGN Acc. | FGSM Acc. | $\theta \in [0, 90°]$ |
|---|---|---|---|---|
| GD | $96.94 \pm 0.08$ | $90.08 \pm 0.07$ | $66.96 \pm 0.49$ | $49.04 \pm 0.46$ |
| GD + WD | $96.93 \pm 0.05$ | $91.93 \pm 0.14$ | $74.20 \pm 0.35$ | $40.83 \pm 0.46$ |
| BNGD Affine F | $97.75 \pm 0.03$ | $49.67 \pm 0.18$ | $0.15 \pm 0.02$ | $90.00 \pm 0.00$ |
| BNGD Affine T | $97.40 \pm 0.07$ | $49.50 \pm 0.20$ | $0.13 \pm 0.02$ | $90.00 \pm 0.00$ |

parameters: $\sigma_\alpha^2 = (\nu_\alpha^j)^2 + \Sigma_{\alpha\alpha}$. The tilting angle $\theta$ of the batch-normalized decision boundary w.r.t. the one given by $w^*$ (note that the boundary is perpendicular to $w$) is therefore approximately equal to the angle between the datasets before and after normalization. To compute $\theta$, we divide the weights $w$ by $\sqrt{\sigma^2 + c}$, and then normalize $w/\|w\|_2$, such that $\theta = \cos^{-1}(w^\top w^*)$. From this analysis it follows that the order of magnitude of $c$ is important relative to the data variance: if $c > \sigma_\alpha^2$ then the effective weight value $w_\alpha$ is reduced, and if $c < \sigma_\alpha^2$ and $\sigma_\alpha^2$ is small, then $w_\alpha$ increases greatly, causing boundary tilting along direction $\alpha$.

We depict simulations of the toy model in Figure 1. We use constant learning rate GD, which is known to converge to the max-margin solution—equivalent to the nearest centroid classifier in this case–for linear models on separable data (Soudry et al., 2018). Batch-normalized GD (BNGD) converges for arbitrary learning rates for linear models (Cai et al., 2019); we use a value of 0.1 for 1000 epochs.

Next, we train linear models on the MNIST 3 vs. 7 dataset with 5000 training samples (drawn uniformly per class) using a learning rate of 0.1 for 50 epochs. We compute the angle $\theta$ w.r.t. the nearest-centroid classifier, which is obtained by subtracting the "average 3" from the "average 7" of the full training set. Although this may seem like a crude reference point, the nearest-centroid classifier is much more robust than the linear model of Goodfellow et al. (2015), achieving $40\%$ accuracy for the fast gradient sign method (FGSM) at $\epsilon = 1/4$ vs. $\approx 0\%$. Results consistent with the boundary tilting theory are shown in Table 1, which not only shows that BN causes tilting, but that this is unaffected by the parameters $\gamma$ and $\beta$. Post-normalization, there is no signal to $\gamma$ and $\beta$ about the variances of the original dataset. This is consistent with other works that observe $\gamma$ and $\beta$ do not influence the studied effect (van Laarhoven, 2017; Zhang et al., 2019a; Yang et al., 2019)

Increasing the numerical stability constant $c$ increases robustness in terms of absolute test accuracy for additive white Gaussian noise (AWGN) on MNIST and CIFAR-10 datasets by $33\%$ and $41\%$ respectively (at the cost of standard accuracy). We defer the details of this experiment to Appendix A. This loss of accuracy, and the effect of $c$ are consistent with Labatie (2019), which remarks that under BN, *directions of high signal variance are dampened, while directions of low signal variance are amplified. This preferential exploration of low signal directions reduces the signal-to-noise ratio and increases sensitivity w.r.t. the input.* Increasing $c$ reduces the sensitivity along "low signal directions".

## 4 EMPIRICAL RESULTS

For the main practical results on MNIST, SVHN, CIFAR-10, and ImageNet, we evaluate the robustness—measured as a drop in test accuracy under various input perturbations—of convolutional networks with and without BN.[1]

As a white-box adversarial attack we use projected gradient descent (PGD) in $\ell_\infty$- and $\ell_2$-norm variants, for its simplicity and ability to degrade performance with little change to the input (Madry et al., 2018). The PGD implementation details are provided in Appendix B. We report the test

---

[1]Unless stated otherwise, we leave the internal parameters of BN to their default values as in modern deep learning frameworks, $c =$1e-5 with $\gamma$ and $\beta$ enabled.

Table 2: (Small learning rate) Test accuracies of VGG8 and WideResNet–28–10 on CIFAR-10 and CIFAR-10.1 (`v6`) in several variants: clean, noisy, and PGD perturbed. We evaluate models achieving the highest validation accuracy after training for 150 epochs.

| | | CIFAR-10 | | | | CIFAR-10.1 | |
|---|---|---|---|---|---|---|---|
| Model | BN | Clean | Noise | PGD-$\ell_\infty$ | PGD-$\ell_2$ | Clean | Noise |
| VGG | ✗ | $87.9 \pm 0.1$ | $79 \pm 1$ | $52.9 \pm 0.6$ | $65.6 \pm 0.3$ | $75.3 \pm 0.2$ | $66 \pm 1$ |
| VGG | ✓ | $88.7 \pm 0.1$ | $73 \pm 1$ | $35.7 \pm 0.3$ | $59.7 \pm 0.3$ | $77.3 \pm 0.2$ | $60 \pm 2$ |
| WRN | F | $94.6 \pm 0.1$ | $69 \pm 1$ | $20.3 \pm 0.3$ | $9.4 \pm 0.2$ | $87.5 \pm 0.3$ | $68 \pm 1$ |
| WRN | ✓ | $95.9 \pm 0.1$ | $58 \pm 2$ | $14.9 \pm 0.6$ | $8.3 \pm 0.3$ | $89.6 \pm 0.2$ | $58 \pm 1$ |

accuracy for additive Gaussian noise of zero mean and variance $1/16$, denoted as "Noise", as well as using the `CIFAR-10-C` common corruption benchmark (Hendrycks & Dietterich, 2019). We found these methods were sufficient to demonstrate a considerable disparity in robustness due to BN, but this is not intended as a complete security evaluation.

Standard meta-parameters from the literature were used to train models with and without BN from scratch over several random seeds. All uncertainties are the standard error of the mean accuracy.[2] For SVHN and CIFAR-10, we examine two different learning rate schemes, given that it has been suggested that one of the primary mechanisms of BN is to facilitate training with a larger learning rate (Ioffe & Szegedy, 2015; Bjorck et al., 2018):

1. A fixed "small" learning rate of 1e-2 (SVHN, CIFAR-10).
2. An initial "large" learning rate of 1e-1, with subsequent drops by a factor of ten (CIFAR-10).

In the SVHN experiment, VGG variants (Simonyan & Zisserman, 2015) are trained using using five random seeds. BN increased clean test accuracy by $1.86 \pm 0.05\%$, but reduced test accuracy for additive noise by $5.5 \pm 0.6\%$, for PGD-$\ell_\infty$ by $17 \pm 1\%$, and for PGD-$\ell_2$ by $20 \pm 1\%$. We defer the full meta-parameters and results to Appendix E.

For the CIFAR-10 experiments we trained models with a similar procedure as for SVHN, but with random $32 \times 32$ crops using four-pixel padding, and horizontal flips. We evaluate two families of contemporary models: one without skip connections (VGG) and a WideResNets (WRN) using "Fixup" initialization (Zhang et al., 2019b) to reduce the use of BN.

Results of the first experiment are summarized in Table 2. In this case, inclusion of BN for VGG reduces the clean generalization gap (difference between training and test accuracy) by $1.1 \pm 0.2\%$. For additive noise, test accuracy drops by $6 \pm 1\%$, and for PGD perturbations by $17.3 \pm 0.7\%$ and $5.9 \pm 0.4\%$ for $\ell_\infty$ and $\ell_2$ variants, respectively.

Very similar results are obtained on a new test set, CIFAR-10.1 `v6` (Recht et al., 2018): BN slightly improves the clean test accuracy (by $2.0 \pm 0.3\%$), but leads to a considerable drop in test accuracy of $6 \pm 1\%$ for the case with additive noise, and $15 \pm 1\%$ and $3.4 \pm 0.6\%$ respectively for $\ell_\infty$ and $\ell_2$ PGD variants (PGD absolute values omitted for CIFAR-10.1 in Table 2 for brevity).

In the second "large" learning rate experiment summarize in Table 3, we prolong training for up to 350 epochs, and drop the learning rate at epoch 150 and 250 in both cases. This increases clean test accuracy relative to results in Table 2. The deepest model that could be trained without BN using this

Table 3: (Large initial learning rate) Robustness of VGG models of increasing depth on CIFAR-10, with and without BN. See text for meta-parameters.

| Model | | Test Accuracy (%) | | |
|---|---|---|---|---|
| L | BN | Clean | Noise | PGD-$\ell_\infty$ |
| 8 | ✗ | $89.29 \pm 0.09$ | $81.7 \pm 0.3$ | $55.6 \pm 0.4$ |
| 8 | ✓ | $90.49 \pm 0.01$ | $77 \pm 1$ | $40.6 \pm 0.6$ |
| 13 | ✗ | $91.74 \pm 0.02$ | $77.8 \pm 0.7$ | $40.3 \pm 0.7$ |
| 13 | ✓ | $93.0 \pm 0.1$ | $67 \pm 1$ | $28.5 \pm 0.4$ |
| 16 | ✓ | $92.8 \pm 0.1$ | $66 \pm 2$ | $28.9 \pm 0.2$ |
| 19 | ✓ | $92.65 \pm 0.09$ | $68 \pm 2$ | $30.0 \pm 0.1$ |

---

[2]Each experiment has a unique uncertainty, hence the number of decimal places varies.

Table 4: Robustness of three modern convolutional neural network architectures with and without BN on the `CIFAR-10-C` common "noise" corruptions (Hendrycks & Dietterich, 2019). We use "F" to denote the Fixup variant of WRN. Values were averaged over five intensity levels for each corruption.

| Model | | | Test Accuracy (%) | | | |
|---|---|---|---|---|---|---|
| Variant | BN | Clean | Gaussian | Impulse | Shot | Speckle |
| VGG8 | ✗ | $87.9 \pm 0.1$ | $\mathbf{65.6 \pm 1.2}$ | $\mathbf{58.8 \pm 0.8}$ | $\mathbf{71.0 \pm 1.2}$ | $\mathbf{70.8 \pm 1.2}$ |
| | ✓ | $88.7 \pm 0.1$ | $56.4 \pm 1.5$ | $51.2 \pm 0.1$ | $65.4 \pm 1.1$ | $66.3 \pm 1.1$ |
| VGG13 | ✗ | $91.74 \pm 0.02$ | $\mathbf{64.5 \pm 0.8}$ | $\mathbf{63.3 \pm 0.3}$ | $\mathbf{70.9 \pm 0.4}$ | $\mathbf{71.5 \pm 0.5}$ |
| | ✓ | $93.0 \pm 0.1$ | $43.6 \pm 1.2$ | $49.7 \pm 0.5$ | $56.8 \pm 0.9$ | $60.4 \pm 0.7$ |
| WRN28 | F | $94.6 \pm 0.1$ | $\mathbf{63.3 \pm 0.9}$ | $\mathbf{66.7 \pm 0.9}$ | $\mathbf{71.7 \pm 0.7}$ | $\mathbf{73.5 \pm 0.6}$ |
| | ✓ | $95.9 \pm 0.1$ | $51.2 \pm 2.7$ | $56.0 \pm 2.7$ | $63.0 \pm 2.5$ | $66.6 \pm 2.5$ |

procedure and He et al. (2015) initialization was VGG13. [3] None of the deeper batch-normalized models recover the robustness of the most shallow, or same-depth equivalents, nor does the higher learning rate recover the performance of the baselines. Results for deeper models are in Appendix E. Other learning rate schedules, and robustness vs. training epoch curves are in Appendix K.

Next, we evaluate robustness on the common corruption benchmark comprising 19 types of real-world distortions from four high-level categories: "noise", "blur", "weather", and "digital" effects (Hendrycks & Dietterich, 2019). Each corruption has five "severity" or intensity levels. We report the mean error on the corrupted test set (mCE) by averaging over all intensity levels and corruptions. We summarize the results for two VGG variants and a WideResNet on `CIFAR-10-C`, trained from scratch on the default training set for three and five random seeds, respectively. Accuracy for the "noise" corruptions, causing the largest difference in accuracy with BN, are outlined in Table 4.

The key takeaway is: *For all models tested, the batch-normalized variant has a higher error rate for all corruptions of the "noise" category, at every intensity level.*

Averaging over all 19 corruptions, BN increases mCE by $1.9 \pm 0.9\%$ for VGG8, $2.0 \pm 0.3\%$ for VGG13, and $1.6 \pm 0.4\%$ for WRN. There is a large disparity in accuracy when modulating BN for different corruption categories, we examine these in more detail in Appendix G.

Interestingly, some corruptions that led to a positive gap for VGG8 show a negative gap for the WRN, i.e., BN improved accuracy to: Contrast— $4.9 \pm 1.1\%$, Snow—$2.8 \pm 0.4\%$, Spatter—$2.3 \pm 0.8\%$. These are the same corruptions for which VGG13 loses, or does not improve its robustness when BN is removed. We suspect accuracy for these corruptions correlates with standard test accuracy, which is highest for the WRN. Visually, these corruptions appear to preserve texture information. Conversely, noise is applied in a spatially global way that disproportionately degrades these textures, emphasizing shapes and edges. It is now known that modern CNNs trained on standard image datasets have a propensity to rely heavily on texture in addition to shape and edge cues for object recognition (Geirhos et al., 2019). We evaluate pre-

Table 5: Robustness of pre-trained ImageNet models with and without BN. *Note*: The numeric suffix indicates number of layers, or the spatial patch width in pixels (of 224) for BagNet.

| | | Top 5 Test Accuracy (%) | | |
|---|---|---|---|---|
| Model | BN | Clean | Noise | PGD-$\ell_\infty$ |
| VGG11 | ✗ | 88.63 | 49.16 | 37.12 |
| VGG11 | ✓ | 89.81 | 49.95 | 26.12 |
| VGG19 | ✗ | 90.88 | 64.86 | 34.19 |
| VGG19 | ✓ | 91.84 | 68.79 | 24.49 |
| AlexNet | ✗ | 79.07 | 41.41 | 39.12 |
| ResNet18 | ✓ | 88.65 | 79.62 | 31.07 |
| BagNet-9 | ✓ | 70.39 | 1.25 | 7.42 |
| BagNet-17 | ✓ | 81.16 | 5.09 | 16.66 |
| BagNet-33 | ✓ | 86.99 | 14.62 | 24.34 |

---

[3]For which one of ten random seeds failed to achieve better than chance accuracy on the training set, while others performed as expected. We report the first three successful runs for consistency with the other experiments.

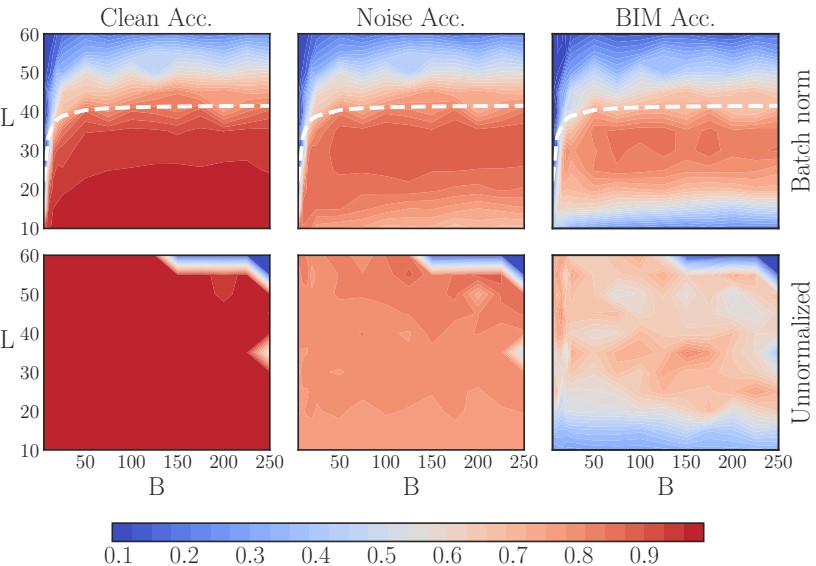

Figure 2: We extend the experiment of Yang et al. (2019) by training fully-connected nets of depth $L$ and constant-width ($N_l = 384$) ReLU layers by SGD, batch size $B$, and learning rate $\eta = 10^{-5}B$ on MNIST. The BN parameters $\gamma$ and $\beta$ were left as default, momentum disabled, and $c = 10^{-3}$. The dashed line is the theoretical maximum trainable depth of batch-normalized networks as a function of the batch size. We report the clean test accuracy, and that for additive Gaussian noise and BIM perturbations. The batch-normalized models were trained for 10 epochs, while the unnormalized ones were trained for 40 epochs as they took longer to converge. The 40 epoch batch-normalized plot was qualitatively similar with dark blue bands for BIM for shallow and deep variants. The dark blue patch for 55 and 60 layer unnormalized models at large batch sizes depicts a total failure to train. These networks were trainable by reducing $\eta$, but for consistency we keep $\eta$ the same in both cases.

trained bag-of-local-feature models (BagNets) on ImageNet with an architecture that discards spatial information between patches and is thus considered to make extensive use of texture patterns for classification (Brendel & Bethge, 2019). For patch sizes $\{9, 17, 33\}$, the top-5 accuracies of the BagNets are reduced to just $1.25\%$, $5.09\%$, and $14.62\%$ for AWGN, respectively. Compared with Table 5, where all models obtain over $40\%$, these figures suggest that robustness to Gaussian noise is a good proxy for the use of texture for ImageNet classification. These results support the hypothesis that BN may be exacerbating this tendency to use superficial texture-like information.

Next, we evaluate the robustness of pre-trained ImageNet models from the `torchvision.models` repository, which conveniently provides models with and without BN.[4] Results are shown in Table 5, where BN improves top-5 accuracy on noise in some cases, but consistently reduces it by $8.54\%$ to $11.00\%$ (absolute) for PGD. The trends are the same for top-1 accuracy, only the absolute values are smaller; the degradation varies from $2.38\%$ to $4.17\%$. Given the discrepancy between noise and PGD for ImageNet, we include a black-box transfer analysis in the Appendix E.2 that is consistent with the white-box analysis.

Finally, we explore the role of batch size and depth in Figure 2. We find that BN limits the maximum trainable depth, which *increases* with the batch size, but quickly plateaus as predicted by Theorem 3.10 of (Yang et al., 2019). Robustness *decreases* with the batch size for depths that maintain a reasonable test accuracy, at around 25 or fewer layers. This tension between clean accuracy and robustness as a function of the batch size is not observed in unnormalized networks.

---

[4]https://pytorch.org/docs/stable/torchvision/models.html, v1.1.0.

## 5    VULNERABILITY AND INPUT DIMENSION

A recent work Simon-Gabriel et al. (2019) analyzes adversarial vulnerability of batch-normalized networks at initialization time and conjectures based on a scaling analysis that, under the commonly used He et al. (2015) initialization scheme, adversarial vulnerability scales as $\sim \sqrt{d}$. They also show in experiments that independence between vulnerability and the input dimension can be approximately recovered through adversarial training by projected gradient descent (PGD) (Madry et al., 2018), with a modest trade-off of clean accuracy. Intuitively, the input dimension should be irrelevant as it does not affect the image complexity (Shafahi et al., 2019).

We show that this can be achieved by simpler means and with little to no trade-off through $\ell_2$ weight decay, where the regularization constant $\lambda$ corrects the loss scaling as the norm of the input increases with $d$. We increase the MNIST image width $\sqrt{d}$ from 28 to 56, 84, and 112 pixels. The loss $\mathcal{L}$ is predicted to grow like $\sqrt{d}$ for $\epsilon$-sized attacks by Thm. 4 of Simon-Gabriel et al. (2019). We confirm that without regularization the loss does scale roughly as predicted (see Table 13 of Appendix F). Training with $\ell_2$ weight decay, however, we obtain adversarial test accuracy ratios of $0.98 \pm 0.01$, $0.96 \pm 0.04$, and $1.00 \pm 0.03$ and clean accuracy ratios of $0.999 \pm 0.002$, $0.996 \pm 0.003$, and $0.987 \pm 0.004$ for $\sqrt{d}$ of 56, 84, and 112, respectively, relative to the original $\sqrt{d} = 28$ dataset. A more detailed explanation and results are provided in Appendix F.

Table 6: Evaluating the robustness of a MLP with and without batch norm. See text for architecture. We observe a $61 \pm 1\%$ reduction in test accuracy due to batch norm for $\sqrt{d} = 84$ compared to $\sqrt{d} = 28$.

| Model | | Test Accuracy (%) | | |
|---|---|---|---|---|
| $\sqrt{d}$ | BN | Clean | Noise | $\epsilon = 0.1$ |
| 28 | ✗ | $97.95 \pm 0.08$ | $93.0 \pm 0.4$ | $66.7 \pm 0.9$ |
| | ✓ | $97.88 \pm 0.09$ | $76.6 \pm 0.7$ | $22.9 \pm 0.7$ |
| 56 | ✗ | $98.19 \pm 0.04$ | $93.8 \pm 0.1$ | $53.2 \pm 0.7$ |
| | ✓ | $98.22 \pm 0.02$ | $79.3 \pm 0.6$ | $8.6 \pm 0.8$ |
| 84 | ✗ | $98.27 \pm 0.04$ | $94.3 \pm 0.1$ | $47.6 \pm 0.8$ |
| | ✓ | $98.28 \pm 0.05$ | $80.5 \pm 0.6$ | $6.1 \pm 0.5$ |

Next, we repeat this experiment with a two-hidden-layer ReLU MLP, with the number of hidden units equal to the half the input dimension, and optionally use one hidden layer with batch norm.[5] To evaluate robustness, 100 iterations of BIM-$\ell_\infty$ were used with a step size of 1e-3, and $\epsilon_\infty = 0.1$. We also report test accuracy with additive Gaussian noise of zero mean and unit variance, the same first two moments as the clean images.[6]

Despite a difference in clean accuracy of only $0.08 \pm 0.05\%$, Table 6 shows that for the original image resolution, batch norm reduced accuracy for noise by $16.4 \pm 0.4\%$, and for BIM-$\ell_\infty$ by $43.8 \pm 0.5\%$. Robustness keeps decreasing as the image size increases,

Table 7: Evaluating the robustness of a MLP with $\ell_2$ weight decay (same $\lambda$ as for linear model, see Table 13 of Appendix F). See text for architecture. Adding batch norm degrades all accuracies.

| Model | | Test Accuracy (%) | | |
|---|---|---|---|---|
| $\sqrt{d}$ | BN | Clean | Noise | $\epsilon = 0.1$ |
| 56 | ✗ | $97.62 \pm 0.06$ | $95.93 \pm 0.06$ | $87.9 \pm 0.2$ |
| | ✓ | $96.23 \pm 0.03$ | $90.22 \pm 0.18$ | $66.2 \pm 0.8$ |
| 84 | ✗ | $96.99 \pm 0.05$ | $95.69 \pm 0.09$ | $87.9 \pm 0.1$ |
| | ✓ | $93.30 \pm 0.09$ | $87.72 \pm 0.11$ | $65.1 \pm 0.5$ |

with the batch-normalized network having $\sim 40\%$ less robustness to BIM and $13 - 16\%$ less to noise at all sizes.

We then apply the $\ell_2$ regularization constants tuned for the respective input dimensions on the linear model to the ReLU MLP with no further adjustments. Table 7 shows that by adding sufficient $\ell_2$ regularization ($\lambda = 0.01$) to recover the original ($\sqrt{d} = 28$, no BN) accuracy for BIM of $\approx 66\%$ when using batch norm, we induce a test error increase of $1.69 \pm 0.01\%$, which is substantial on

---

[5]This choice of architecture is mostly arbitrary, the trends were the same for constant width layers.

[6]We first apply the noise to the original 28×28 pixel images, then resize them to preserve the appearance of the noise.

MNIST. Furthermore, using the same regularization constant and no batch norm increases clean test accuracy by $1.39 \pm 0.04\%$, and for the BIM-$\ell_\infty$ perturbation by $21.7 \pm 0.4\%$.

Finally, following the guidance in the original work on batch norm (Ioffe & Szegedy, 2015) to the extreme ($\lambda = 0$): when we *reduce* weight decay when using batch norm, accuracy for the $\epsilon_\infty = 0.1$ perturbation is degraded by $79.3 \pm 0.3\%$ for $\sqrt{d} = 56$, and $81.2 \pm 0.2\%$ for $\sqrt{d} = 84$.

*In all cases, using batch norm greatly reduced test accuracy for noisy and adversarially perturbed inputs, while weight decay increased accuracy for such inputs.*

As supplementary evidence, we contrast the "Fooling images" (Nguyen et al., 2015) and Carlini & Wagner (2017) examples of BN vs. L2-regularized models on MNIST and SVHN in Appendix J.

## 6 RELATED WORK

Our work examines the effect of batch norm on model robustness at test time. References with an immediate connection to our work were discussed in the previous sections; here we briefly mention other works that do not have a direct relationship to our experiments, but are relevant to batch norm in general.

The original work Ioffe & Szegedy (2015) that introduced batch norm as a technique for improving neural network training and test performance motivated it in terms of "internal covariate shift" – referring to the changing distribution of layer outputs, an effect that requires subsequent layers to steadily adapt to the new distribution and thus slows down the training process. Several follow-up works started from the empirical observation that batch norm usually accelerates and stabilizes training, and attempted to clarify the mechanism behind this effect. One argument is that batch-normalized networks have a smoother optimization landscape due to smaller gradients immediately before the batch-normalized layer (Santurkar et al., 2018). However, Yang et al. (2019) study the effect of stacking many batch-normalized layers and prove that this causes gradient explosion that is exponential in network depth for networks without skip connections and holds for any non-linearity. In practice, relatively shallow batch-normalized networks seem to benefit from the "helpful smoothing" of the loss surface property Santurkar et al. (2018), while very deep networks are not trainable (Yang et al., 2019). In our work, we found that a single batch-normalized layer suffices to induce severe adversarial vulnerability.

A concurrent submission suggests that BN induced vulnerability may be due to a mismatch between the tracked mean and variance values at training versus test time (Anonymous, 2020). We investigate this hypothesis in Appendix I and find that the use of tracked statistics can play a similar role as tuning the numerical stability constant $c$, thus this does not completely account for the vulnerability.

Weight decay's loss scaling mechanism is complementary to other mechanisms identified in the literature, for instance that it increases the effective learning rate (van Laarhoven, 2017; Zhang et al., 2019a). Our results are consistent with these works in that weight decay reduces the generalization gap (between training and test error), even in batch-normalized networks where it is presumed to have no effect. Given that batch norm is not typically used on all layers, the loss scaling mechanism persists, although to a lesser degree in this case.

Shafahi et al. (2019) performed similar input dimension scaling experiments as in this work and came to a similar conclusion that the input dimension is irrelevant to adversarial vulnerability. However, like Simon-Gabriel et al. (2019), they use PGD rather than weight decay to prevent vulnerability from increasing with input dimension. Although it can be shown that robust optimization is equivalent to parameter norm regularization for linear models if we allow the $\epsilon$-ball (aka disturbance $\delta$) to vary with each example (Xu et al., 2009), we maintain that the latter is a more efficient approach.

## 7 CONCLUSION

We found that there is no free lunch with batch norm when model robustness is a concern: the accelerated training properties and occasionally higher clean test accuracy come at the cost of increased vulnerability, both to additive noise and for adversarial perturbations. We have shown that there is no inherent relationship between the input dimension and vulnerability. Our results highlight the importance of identifying the disparate mechanisms of regularization techniques.

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

## A  THE NUMERICAL STABILITY CONSTANT

The constant $c$ originally added to the mini-batch variance in the denominator for numerical stability (named $\epsilon$ in Ioffe & Szegedy (2015)) turns out to be an important hyperparameter in terms of robustness. It acts as a threshold on the variance of all input dimensions or neurons. When $c$ is much less than the minimum variance over dimensions, it induces boundary tilting along the low-variance dimensions. In Figure 3 we sweep $c$ for MNIST 3 vs. 7 and CIFAR-10, and compare the corresponding clean test accuracy with FGSM and AWGN accuracy for MNIST, and AWGN for CIFAR-10. For MNIST, increasing $c$ allows us to trade-off clean accuracy for robustness to FGSM, but is suboptimal compared to L2 weight decay. For these experiments we fixed $\gamma_\alpha = 1$ and $\beta_\alpha = 0$.

For CIFAR-10, eight-layer VGG models were trained with a constant learning rate of 0.01 with no drops, momentum of 0.9, a batch size of 128, and 50 epochs (for computational reasons) over four random seeds. As for BNGD, for this particular experiment we apply BN only to the input layer. A consistent trend is observed where robustness to noise increases greatly as $c$ is increased, but we note that this occurs for $c$ several orders of magnitude greater than default settings.

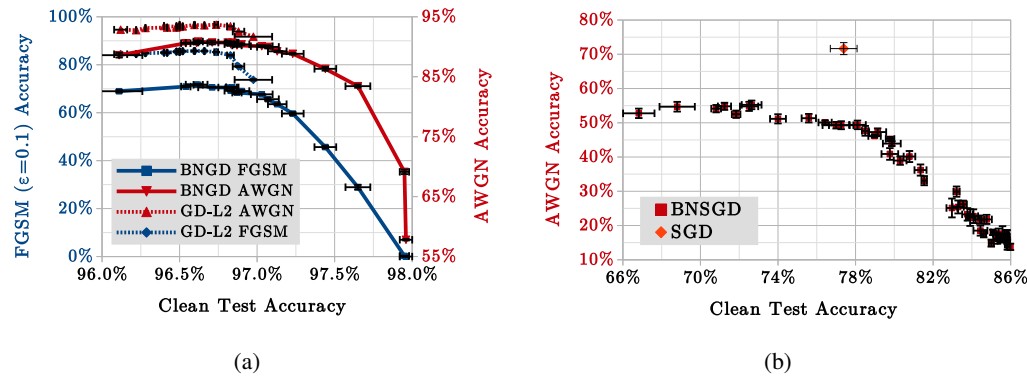

(a)                                                                 (b)

Figure 3: Sweeping the BN numerical stability constant for (a) the MNIST 3 vs. 7 dataset with $c \in$ [1e-3, 2e+1] for BNGD, and as a baseline the L2 regularization constant $\lambda \in$ [1e-3, 9] (shown as "GD-L2"). (b) Sweeping $c$ for batch-normalized SGD (BNSGD) on the CIFAR-10 dataset with $c \in$ [1e-6, 3e+3] for the VGG8 architecture. Increasing either $c$ or $\lambda$ has a similar effect to trade clean test accuracy for increased robustness, until the effect is too large and both accuracies degrade. The absolute accuracies are consistently higher without BN. Error bars indicate standard error of the mean over four and five random seeds for MNIST and CIFAR-10, respectively. We recommend following each curve from right to left to be consistent with our description above. The default setting (highest clean test accuracy, lowest robustness) starts in the bottom right corner and the initial trade-off between clean test accuracy and robustness is traced up and leftwards until the curves inflect.

## B    PGD IMPLEMENTATION DETAILS

We used the PGD implementation from (Ding et al., 2019), with an example code snippet below. Because we normalize the data to zero mean and unit variance for SVHN, CIFAR-10, and ImageNet, the pixel range was clipped to $\{\pm 1\}$, $\{\pm 2\}$, and $\{\pm 2\}$ respectively.[7] These ranges were obtained by visual inspection of the image histograms and confirming that most of the pixels fall within this range. As a sanity check, we set the perturbation magnitude $\epsilon = 0$ to confirm that the clipping itself had a negligible impact on test accuracy.

As a result, absolute accuracy comparisons with other works where the input is not centered, e.g. $x \in$ [0, 1], should be compared to $\epsilon = 4/255$ in those works for $\epsilon = 8/255$ in this work.

### B.1    FOR EVALUATION

During the evaluation of robustness with PGD, we use $\epsilon_\infty = 0.03$ and a step size of $\epsilon_\infty/10$ for SVHN, CIFAR-10, and $\epsilon_\infty = 0.01$ for ImageNet. For PGD-$\ell_2$ we set $\epsilon_2 = \epsilon_\infty \sqrt{d}$, where $d$ is the input dimension. To reduce the random error of the evaluation no random start was used at test-time, i.e. `rand_init=False`. A code snippet for the PGD-$\ell_\infty$ evaluation on SVHN is shown below:

```
from advertorch.attacks import LinfPGDAttack
adversary = LinfPGDAttack(net, loss_fn=nn.CrossEntropyLoss(reduction="sum"),
    eps=0.03, nb_iter=20, eps_iter=0.003,
    rand_init=False, clip_min=-1.0, clip_max=1.0, targeted=False)
```

We report 20 iterations in most cases unless stated otherwise, but confirmed that using 40 iterations did not significantly improve upon this to within the measurement random error, i.e., the accuracy was not degraded further. For example, for VGG16 on CIFAR-10 evaluated using 40 iterations of PGD with $\epsilon_\infty = 0.03$ and a step size of $\epsilon_\infty/20$, instead of 20 iterations with $\epsilon_\infty/10$, accuracy changed from $28.9 \pm 0.2\%$ to $28.5 \pm 0.3\%$, which is a difference of only $0.4 \pm 0.5\%$, less than the random (standard) error.

---

[7]For CIFAR-10, see e.g. https://github.com/kuangliu/pytorch-cifar/blob/master/main.py#L33 for the preprocessing scheme we used.

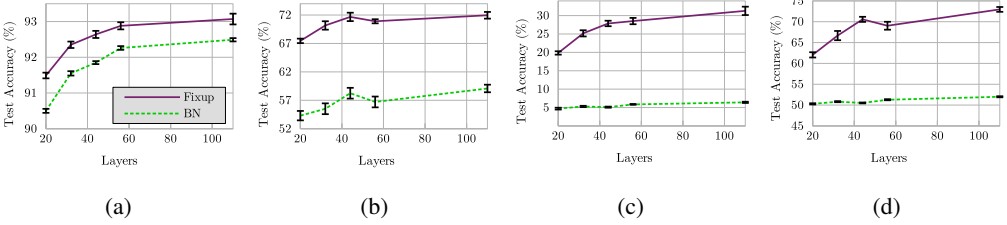

Figure 4: Accuracy of batch-normalized versus unnormalized (Fixup) residual networks of varying depth. Models were trained with standard hyperparameters and evaluated on CIFAR-10: (a) clean test set, (b) noisy test set, (c) PGD $\ell_\infty$, and (d) PGD $\ell_2$. Error bars denote the standard error of the mean over five random seeds. The unnormalized networks obtain higher accuracy for all tests and depths.

### B.2 FOR PGD ADVERSARIAL TRAINING

For PGD *training*, the initial random perturbation (`rand_init=True`) was used to improve the diversity of adversarial examples generated as per Madry et al. (2018). The meta-parameters are provided on a case-by-case basis in Appendix D.

## C ON AN ACCURACY VS. ROBUSTNESS TRADE-OFF

It is natural to wonder if the degradation in robustness arising from the use of BN is simply due to BN increasing the standard test accuracy, given a known trade-off between the two (Tanay & Griffin, 2016; Galloway et al., 2018; Su et al., 2018; Tsipras et al., 2019). Note that if the relationship between input $X$ and label $Y$ is free of noise, e.g., as in Gilmer et al. (2018), then there is no such trade-off and increasing accuracy corresponds to increasing robustness. For the toy problem we studied in § 3, BN actually aligned the decision boundary with the Bayes-optimal solution, so increasing standard accuracy may be intrinsic to the normalization itself in some cases.

Given that BN does typically increase clean test accuracy by some small amount on commonly used datasets, we thought it was most representative to not intentionally limit the performance of BN. We did, however, find natural cases where BN did *not* improve clean test accuracy. We trained ResNets{20,32,44,56,110} using Fixup initialization on CIFAR-10: all consistently obtain about 0.5% higher clean test accuracy than their batch-normalized equivalent, and are also more robust to noise ($\approx 15\%$) and PGD $\ell_\infty$ and $\ell_2$ perturbations ($\approx 30\%$), as shown in Figure 4.

For MNIST, the results of Tables 6 & 7 also show compatible clean accuracy irrespective of BN, and yet vastly different robustness. Thus, the vulnerability induced by BN is not merely a consequence of increasing standard test accuracy.

## D PGD TRAINING YIELDS UNFORTUNATE ROBUSTNESS TRADE-OFFS

We believe that it is most informative to evaluate on corruptions or perturbations that are not presented to models during training, which is why we did not use PGD adversarial training in the main text given that we evaluate on the same. In a practical setting we cannot assume awareness of potential attacks *a priori*.

For brevity, we opted to report accuracy for an arbitrary small value of $\epsilon$ in the main text. In general, however, it is more informative to plot accuracy vs. $\epsilon$ to ensure the accuracy reaches zero for reasonably large $\epsilon$ to help rule out gradient masking issues (Papernot et al., 2017; Athalye et al., 2018). This also shows that $\epsilon$ was not cherry-picked. Figure 5(b) shows that PGD-$\ell_\infty$ training recovers much of the BN-vulnerability gap when tested on PGD-$\ell_\infty$ perturbations only, although there is a still a non-trivial improvement at $\epsilon = 8/255$ from 38.84% to 41.57% (recall that we only trained with $\epsilon_{\max} = 4/255$, so absolute accuracy is slightly lower than in Madry et al. (2018)).

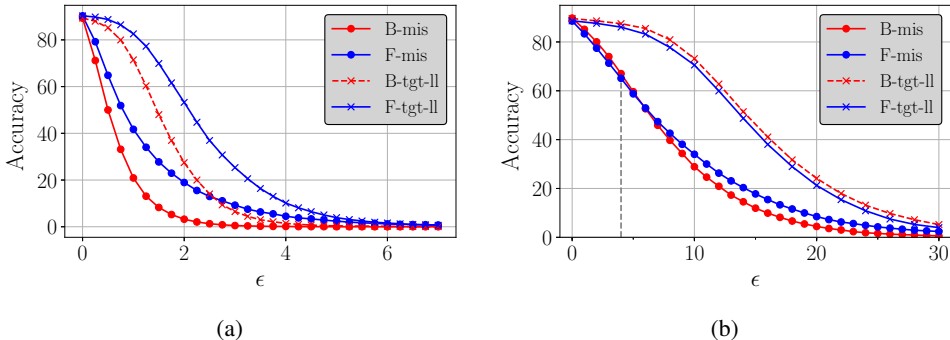

(a)          (b)

Figure 5: Test accuracy vs. $\epsilon$ for (a), a naturally trained ResNet32 and (b)a state-of-the-art WideResNet 28-10 CIFAR-10 baseline (Madry et al., 2018) trained with PGD-$\ell_\infty$ ($\epsilon_{\max}$ = 4, 5 iterations, step size 1 out of 255, w/rand. start) in batch-normalized ('B') and unnormalized ('F' for Fixup) variants. At test-time, 20 PGD iterations are used (the "strong" adversary from Madry et al.) for the $\ell_\infty$, and $\ell_2$ norms. A misclassification objective ('mis') is compared to targeting the least-likely label ('tgt-ll').

Ultimately, adversarial robustness is concerned with robustness to the worst attack in our threat model. If we consider the `contrast` corruption from `CIFAR-10-C` (Hendrycks & Dietterich, 2019), PGD-$\ell_\infty$ training with $\epsilon = 4$, 5 iterations, and a step size of 1 (as in Figure 5(b)), this reduces the absolute test accuracy by $23.5\%$ and $28.5\%$ for WideResNet 28–10 in Fixup and BN variants, respectively. We do not believe it is reasonable for a threat model of natural image classifiers to exclude natural changes in image contrast (Gilmer & Hendrycks, 2019). Thus, the recipe of PGD training a high capacity model alone can exacerbate other vulnerabilities (Jacobsen et al., 2019a; Mu & Gilmer, 2019).

Similar results are observed on an `MNIST-C` benchmark (Mu & Gilmer, 2019), to which we add an additional Gaussian noise corruption. We PGD-train the LeNet-5 variant from the `AdverTorch` library (Ding et al., 2019) using 20 inner-loop iterations, a step size of $^{0.3}/_{10}$, and $\epsilon_{\max} = ^{0.3}/_{1.0}$ with a random initial perturbation as in Madry et al. (2018). Table 8 shows that BN reduces the mean test accuracy on `MNIST-C` by $11 \pm 2\%$; the accuracy is reduced from $87.0 \pm 0.5\%$ (Per-img) to $76 \pm 2\%$ (BN). This is compatible with Mu & Gilmer (2019), which reported an $11.15\%$ degradation in absolute test accuracy for PGD training, although they did not report the variance over multiple runs. Note that Mu & Gilmer (2019) use a slightly different CNN architecture that obtains $91.21\%$ mean test accuracy, slightly higher than our $87.0 \pm 0.5\%$ baseline.

In particular, the performance of BN and the PGD variants is decimated by altering the brightness of the images, whereas the baseline shows little performance degradation. It is known that PGD training yields a highly sparse network that implements a thresholding operation in the first layer, see the Appendix C of Madry et al. (2018), which forces the model to be invariant to perturbations with $\ell_\infty$-norm $< \epsilon$. This excessive invariance is itself another cause of adversarial vulnerability (Jacobsen et al., 2019a;b). The PGD trained batch-normalized model (PGD BN) performs worse than the baseline by double digit percentages for each of the: "Brightness", "Fog", "Impulse Noise", "Stripe", and "Zigzag" corruptions, with $7 \pm 2\%$ higher mCE.

Table 8: MNIST-C corruption benchmark results for four models: A naturally trained baseline using per-image standardization, "Per-img", a naturally trained batch-normalized baseline, "BN", a PGD trained model, "PGD", and a PGD trained batch-normalized model, "PGD BN". See text for PGD details. Values are the accuracy after applying each corruption to the test set, and the standard error over three random initializations of each model, to one or two significant figures. Cells for which the "Per-img" baseline dramatically outperforms *all* other columns are emphasized in grey. For "BN", "PGD", and "PGD BN", corruption accuracies can fluctuate considerably with the random seed, despite low variance in clean test accuracy. Conversely, the accuracy of Per-img has low variance in all cases.

| | Model | | | |
|---|---|---|---|---|
| Corruption | Per-img | BN | PGD | PGD BN |
| Clean | $98.50 \pm 0.05\%$ | $99.12 \pm 0.03\%$ | $98.51 \pm 0.05\%$ | $98.86 \pm 0.02\%$ |
| Brightness | $98.46 \pm 0.06\%$ | $48 \pm 12\%$ | $27 \pm 3\%$ | $40 \pm 9\%$ |
| Canny Edges | $77 \pm 2\%$ | $79 \pm 3\%$ | $86 \pm 2\%$ | $84 \pm 2\%$ |
| Dotted Line | $96.7 \pm 0.1\%$ | $95.5 \pm 0.5\%$ | $90 \pm 2\%$ | $90 \pm 1\%$ |
| Fog | $82.0 \pm 0.2\%$ | $23 \pm 5\%$ | $58 \pm 3\%$ | $69 \pm 2\%$ |
| Glass Blur | $91 \pm 1\%$ | $53 \pm 2\%$ | $94.6 \pm 0.3\%$ | $95.0 \pm 0.3\%$ |
| Gaussian Noise | $90 \pm 1\%$ | $84.7 \pm 0.6\%$ | $91 \pm 2\%$ | $94.2 \pm 0.6\%$ |
| Impulse Noise | $96.5 \pm 0.2\%$ | $50 \pm 2\%$ | $50 \pm 4\%$ | $57 \pm 2\%$ |
| Motion Blur | $80.5 \pm 0.6\%$ | $86 \pm 2\%$ | $93.4 \pm 0.2\%$ | $93.6 \pm 0.3\%$ |
| Rotate | $89.1 \pm 0.2\%$ | $91.6 \pm 0.2\%$ | $91.1 \pm 0.3\%$ | $92.5 \pm 0.1\%$ |
| Scale | $87.9 \pm 0.5\%$ | $94.3 \pm 0.2\%$ | $88 \pm 1\%$ | $91.8 \pm 0.5\%$ |
| Shear | $95.2 \pm 0.2\%$ | $97.3 \pm 0.2\%$ | $97.2 \pm 0.1\%$ | $97.63 \pm 0.07\%$ |
| Shot Noise | $97.8 \pm 0.1\%$ | $83.7 \pm 0.8\%$ | $98.2 \pm 0.1\%$ | $98.60 \pm 0.04\%$ |
| Spatter | $90.4 \pm 0.5\%$ | $97.60 \pm 0.08\%$ | $96.2 \pm 0.2\%$ | $96.44 \pm 0.08\%$ |
| Stripe | $85.0 \pm 0.2\%$ | $92 \pm 1\%$ | $52 \pm 3\%$ | $51 \pm 8\%$ |
| Translate | $46.7 \pm 0.5\%$ | $56.6 \pm 0.7\%$ | $51 \pm 1\%$ | $54.6 \pm 0.8\%$ |
| Zigzag | $88.0 \pm 0.8\%$ | $81 \pm 1\%$ | $71 \pm 1\%$ | $76.3 \pm 0.3\%$ |
| **Overall** | $87.0 \pm 0.5\%$ | $76 \pm 2\%$ | $77 \pm 1\%$ | $80 \pm 2\%$ |

Table 10: (Small learning rate) VGG variants on SVHN with BN.

| | Test Accuracy (%) | | | |
|---|---|---|---|---|
| L | Clean | Noise | PGD-$\ell_\infty$ | PGD-$\ell_2$ |
| 11 | $95.31 \pm 0.03$ | $80.5 \pm 1$ | $20.2 \pm 0.2$ | $6.1 \pm 0.2$ |
| 13 | $95.88 \pm 0.05$ | $77.2 \pm 7$ | $21.7 \pm 0.5$ | $5.4 \pm 0.2$ |
| 16 | $94.59 \pm 0.05$ | $78.1 \pm 4$ | $19.2 \pm 0.3$ | $3.0 \pm 0.2$ |
| 19 | $95.1 \pm 0.3$ | $78 \pm 1$ | $24.2 \pm 0.6$ | $4.1 \pm 0.4$ |

Table 11: (Large learning rate) Accuracies of WideResNet–28–10 on CIFAR-10 and CIFAR-10.1 (`v6`).

| | CIFAR-10 | | | | CIFAR-10.1 | |
|---|---|---|---|---|---|---|
| Model | Clean | Noise | PGD-$\ell_\infty$ | PGD-$\ell_2$ | Clean | Noise |
| Fixup | $94.6 \pm 0.1$ | $69.1 \pm 1.1$ | $20.3 \pm 0.3$ | $9.4 \pm 0.2$ | $87.5 \pm 0.3$ | $67.8 \pm 0.9$ |
| BN | $95.9 \pm 0.1$ | $57.6 \pm 1.5$ | $14.9 \pm 0.6$ | $8.3 \pm 0.3$ | $89.6 \pm 0.2$ | $58.3 \pm 1.2$ |

# E    ADDITIONAL EMPIRICAL RESULTS

This section contains supplementary explanations and results to those of Section 4.

## E.1    DETAILED SVHN AND CIFAR-10 RESULTS

For SVHN, models were trained by stochastic gradient descent (SGD) with momentum 0.9 for 50 epochs, with a batch size of 128 and initial learning rate of 0.01, which was dropped by a factor of ten at epochs 25 and 40. Trials were repeated over five random

Table 9: (Small learning rate) Test accuracies of VGG8 on SVHN.

| BN | Clean | Noise | PGD-$\ell_\infty$ | PGD-$\ell_2$ |
|---|---|---|---|---|
| ✗ | $92.60 \pm 0.04$ | $83.6 \pm 0.2$ | $27.1 \pm 0.3$ | $22.0 \pm 0.8$ |
| ✓ | $94.46 \pm 0.02$ | $78.1 \pm 0.6$ | $10 \pm 1$ | $1.6 \pm 0.3$ |

seeds. We show the results of this experiment in Table 9, finding that BN increased clean test accuracy by $1.86 \pm 0.05\%$, and reduced test accuracy for additive noise by $5.5 \pm 0.6\%$, for PGD-$\ell_\infty$ by $17 \pm 1\%$, and for PGD-$\ell_2$ by $20 \pm 1\%$.

Our first attempt to train VGG models on SVHN with more than eight layers failed, therefore for a fair comparison we report the robustness of the deeper models that were only trainable by using BN in Table 10. None of these models obtained much better robustness in terms of PGD-$\ell_2$, although they did better for PGD-$\ell_\infty$.

Fixup initialization was recently proposed to reduce the use of normalization layers in deep residual networks (Zhang et al., 2019b). As a natural test we compare a WideResNet (28 layers, width factor 10) with Fixup versus the default architecture with BN. Note that the Fixup variant still contains one BN layer before the classification layer, but the number of BN layers is still greatly reduced.[8]

We train WideResNets (WRN) with five unique seeds and report their test accuracies in Table 11. Consistent with (Recht et al., 2018), higher clean test accuracy on CIFAR-10, i.e. obtained by the WRN compared to VGG, translated to higher clean accuracy on CIFAR-10.1. However, these gains were wiped out by moderate Gaussian noise. VGG8 dramatically outperforms both WideResNet variants subject to noise, achieving $78.9\pm0.6$ vs. $69.1\pm1.1$. Unlike for VGG8, the WRN showed little generalization gap between noisy CIFAR-10 and 10.1 variants: $69.1 \pm 1.1$ is reasonably comparable

---

[8]We used the implementation from `https://github.com/valilenk/fixup`, but stopped training at 150 epochs for consistency with the VGG8 experiment. Both models had already fit the training set by this point.

Table 12: ImageNet validation accuracy for adversarial examples transfered between VGG variants of various depths, indicated by number, with and without BN ("✓", "✗"). All adversarial examples were crafted with BIM-$\ell_\infty$ using 10 steps and a step size of 5e-3, which is higher than for the white-box analysis to improve transferability. The BIM objective was simply misclassification, i.e., it was not a targeted attack. For efficiency reasons, we select 2048 samples from the validation set. Values along the diagonal in first two columns for Source = Target indicate white-box accuracy.

| | | | Target | | | | | | | |
| --- | --- | --- | --- | --- | --- | --- | --- | --- | --- | --- |
| | | | 11 | | 13 | | 16 | | 19 | |
| Acc. Type | Source | | ✗ | ✓ | ✗ | ✓ | ✗ | ✓ | ✗ | ✓ |
| Top 1 | 11 | ✗ | 1.2 | 42.4 | 37.8 | 42.9 | 43.8 | 49.6 | 47.9 | 53.8 |
| | | ✓ | 58.8 | 0.3 | 58.2 | 45.0 | 61.6 | 54.1 | 64.4 | 58.7 |
| Top 5 | 11 | ✗ | 11.9 | 80.4 | 75.9 | 80.9 | 80.3 | 83.3 | 81.6 | 85.1 |
| | | ✓ | 87.9 | 6.8 | 86.7 | 83.7 | 89.0 | 85.7 | 90.4 | 88.1 |

with $67.8 \pm 0.9$, and $57.6 \pm 1.5$ with $58.3 \pm 1.2$. The Fixup variant improves accuracy by $11.6 \pm 1.9\%$ for noisy CIFAR-10, $9.5 \pm 1.5\%$ for noisy CIFAR-10.1, $5.4 \pm 0.6\%$ for PGD-$\ell_\infty$, and $1.1 \pm 0.4\%$ for PGD-$\ell_2$.

We believe our work serves as a compelling motivation for Fixup and other techniques that aim to reduce usage of BN. The role of skip-connections should be isolated in future work since absolute values were consistently lower for residual networks.

### E.2 IMAGENET BLACK-BOX TRANSFERABILITY ANALYSIS

The discrepancy between the results in additive noise and for white-box BIM perturbations for ImageNet in Section 3 raises a natural question: Is *gradient masking* a factor influencing the success of the white-box results on ImageNet? No, consistent with the white-box results, when the target is unnormalized but the source is, top 1 accuracy is $10.5\% - 16.4\%$ higher, while top 5 accuracy is $5.3\% - 7.5\%$ higher, than vice versa. This can be observed in Table 12 by comparing the diagonals from lower left to upper right. When targeting an unnormalized model, we reduce top 1 accuracy by $16.5\% - 20.4\%$ using a source that is also unnormalized, compared to a difference of only $2.1\% - 4.9\%$ by matching batch-normalized networks. This suggests that the features used by unnormalized networks are more stable than those of batch-normalized networks.

Unfortunately, the pre-trained ImageNet models provided by the PyTorch developers do not include hyperparameter settings or other training details. However, we believe that this speaks to the generality of the results, i.e., that they are not sensitive to hyperparameters.

### E.3 BATCH NORM LIMITS MAXIMUM TRAINABLE DEPTH AND ROBUSTNESS

In Figure 6 we show that BN not only limits the maximum trainable depth, but robustness decreases with the batch size for depths that maintain test accuracy, at around 25 or fewer layers (in Figure 6(a)). Both clean accuracy and robustness showed little to no relationship with depth nor batch size in unnormalized networks. A few outliers are observed for unnormalized networks at large depths and batch size, which could be due to the reduced number of parameter update steps that result from a higher batch size and fixed number of epochs (Hoffer et al., 2017).

Note that in Figure 6(a) the bottom row—without batch norm—appears lighter than the equivalent plot above, with batch norm, indicating that unnormalized networks obtain less absolute peak accuracy than the batch-normalized network. Given that the unnormalized networks take longer to converge, we prolong training for 40 epochs total. When they do converge, we see more configurations that achieve higher clean test accuracy than batch-normalized networks in Figure 6(b). Furthermore, good robustness can be experienced simultaneously with good clean test accuracy in unnormalized networks, whereas the regimes of good clean accuracy and robustness are still mostly non-overlapping in Figure 6(b).

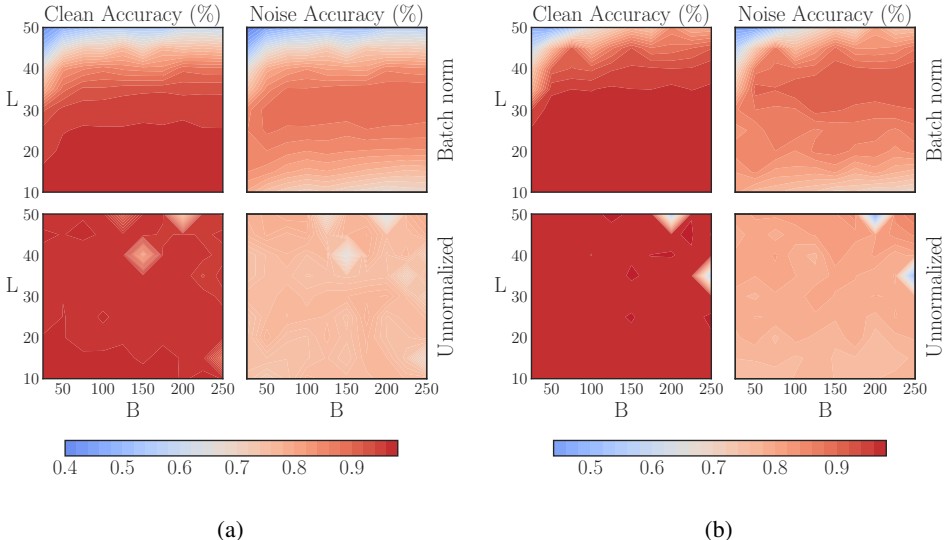

Figure 6: We repeat the experiment of Yang et al. (2019) by training fully-connected models of depth $L$ and constant width ($N_l$=384) with ReLU units by SGD, and learning rate $\eta = 10^{-5}B$ for batch size $B$ on MNIST. We train for 10 and 40 epochs in (a) and (b) respectively. The BN parameters $\gamma$ and $\beta$ were left as default, momentum disabled, and $c = 1e$-3. Each coordinate is first averaged over three seeds. Diamond-shaped artefacts for unnormalized case indicate one of three seeds failed to train – note that we show an equivalent version of (a) with these outliers removed and additional batch sizes from 5–20 in Figure 2. Best viewed in colour.

## F  WEIGHT DECAY AND INPUT DIMENSION

Consider a logistic classification model represented by a neural network consisting of a single unit, parameterized by weights $w \in \mathbb{R}^d$ and bias $b \in \mathbb{R}$, with input denoted by $x \in \mathbb{R}^d$ and true labels $y \in \{\pm 1\}$. Predictions are defined by $s = w^\top x + b$, and the model is optimized through empirical risk minimization, i.e., by applying stochastic gradient descent (SGD) to the loss function equation 2, where $\zeta(z) = \log(1 + e^{-z})$:

$$\mathbb{E}_{x,y \sim p_{\text{data}}} \zeta(y(w^\top x + b)). \tag{2}$$

We note that $w^\top x + b$ is a *scaled*, signed distance between $x$ and the classification boundary defined by our model. If we define $d(x)$ as the signed Euclidean distance between $x$ and the boundary, then we have: $w^\top x + b = \|w\|_2\, d(x)$. Hence, minimizing equation 2 is equivalent to minimizing

$$\mathbb{E}_{x,y \sim p_{\text{data}}} \zeta(\|w\|_2 \times y\, d(x)). \tag{3}$$

We define the *scaled loss* as

$$\zeta_{\|w\|_2}(z) := \zeta(\|w\|_2 \times z) \tag{4}$$

and note that adding a $\ell_2$ regularization term in equation 3, resulting in equation 5, can be understood as a way of controlling the scaling of the loss function:

$$\mathbb{E}_{x,y \sim p_{\text{data}}} \zeta_{\|w\|_2}(y\, d(x)) + \lambda \|w\|_2 \tag{5}$$

In Figures 7(a)-7(c), we develop intuition for the different quantities contained in equation 2 with respect to a typical binary classification problem, while Figures 7(d)-7(f) depict the effect of the regularization parameter $\lambda$ on the scaling of the loss function.

To test this theory empirically we study a model with a single linear layer (number of units equals input dimension) and cross-entropy loss function on variants of MNIST of increasing input dimension, to approximate the toy model described in the "core idea" from (Simon-Gabriel et al., 2019) as closely as possible, but with a model capable of learning. Clearly, this model is too simple to obtain competitive test accuracy, but this is a helpful first step that will be subsequently extended to ReLU

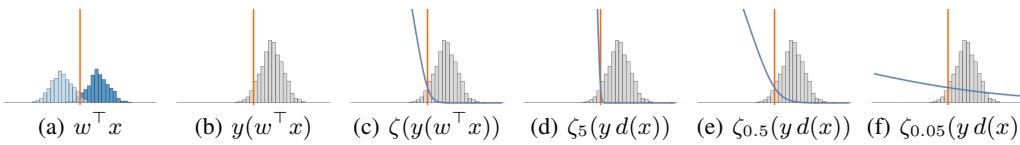

(a) $w^\top x$  (b) $y(w^\top x)$  (c) $\zeta(y(w^\top x))$  (d) $\zeta_5(y\,d(x))$  (e) $\zeta_{0.5}(y\,d(x))$  (f) $\zeta_{0.05}(y\,d(x))$

Figure 7: (a) For a given weight vector $w$ and bias $b$, the values of $w^\top x + b$ over the training set typically follow a bimodal distribution (corresponding to the two classes) centered on the classification boundary. (b) Multiplying by the label $y$ allows us to distinguish the correctly classified data in the positive region from misclassified data in the negative region. (c) We can then attribute a penalty to each training point by applying the loss to $y(w^\top x + b)$. (d) For a small regularization parameter (large $\|w\|_2$), the misclassified data is penalized linearly while the correctly classified data is not penalized. (e) A medium regularization parameter (medium $\|w\|_2$) corresponds to smoothly blending the margin. (f) For a large regularization parameter (small $\|w\|_2$), all data points are penalized almost linearly.

networks. The model was trained by SGD for 50 epochs with a constant learning rate of 1e-2 and a mini-batch size of 128. In Table 13 we show that increasing the input dimension by resizing MNIST from $28 \times 28$ to various resolutions with `PIL.Image.NEAREST` interpolation increases adversarial vulnerability in terms of accuracy and loss. Furthermore, the "adversarial damage", defined as the average increase of the loss after attack, which is predicted to grow like $\sqrt{d}$ by Theorem 4 of (Simon-Gabriel et al., 2019), falls in between that obtained empirically for $\epsilon = 0.05$ and $\epsilon = 0.1$ for all image widths except for 112, which experiences slightly more damage than anticipated.

Simon-Gabriel et al. (2019) note that independence between vulnerability and the input dimension can be recovered through adversarial-example augmented training by projected gradient descent (PGD), with a small trade-off in terms of standard test accuracy. We find that the same can be achieved through a much simpler approach: $\ell_2$ weight decay, with parameter $\lambda$ chosen dependent on $d$ to correct for the loss scaling. This way we recover input dimension invariant vulnerability with little degradation of test accuracy, e.g., see the result for $\sqrt{d} = 112$ and $\epsilon = 0.1$ in Table 13: the accuracy ratio is $1.00 \pm 0.03$ with weight decay regularization, compared to $0.10 \pm 0.09$ without.

Compared to PGD training, weight decay regularization i) does not have an arbitrary $\epsilon$ hyperparameter that ignores inter-sample distances, ii) does not prolong training by a multiplicative factor given by the number of steps in the inner loop, and 3) is less attack-specific. Thus, we do not use adversarially augmented training because we wish to convey a notion of robustness to unseen attacks and common corruptions. Furthermore, enforcing robustness to $\epsilon$-perturbations may increase vulnerability to *invariance-based* examples, where semantic changes are made to the input, thus changing the Oracle label, but not the classifier's prediction (Jacobsen et al., 2019a). Our models trained with weight decay obtained 12% higher accuracy (86% vs. 74% correct) compared to batch norm on a small sample of 100 $\ell_\infty$ invariance-based MNIST examples.[9] We make primary use of traditional $\ell_p$ perturbations as they are well studied in the literature and straightforward to compute, but solely defending against these is not the end goal.

A more detailed comparison between adversarial training and weight decay can be found in (Galloway et al., 2018). The scaling of the loss function mechanism of weight decay is complementary to other mechanisms identified in the literature recently, for instance that it also increases the effective learning rate (van Laarhoven, 2017; Zhang et al., 2019a). Our results are consistent with these works in that weight decay reduces the generalization gap, even in batch-normalized networks where it is presumed to have no effect. Given that batch norm is not typically used on the last layer, the loss scaling mechanism persists in this setting, albeit to a lesser degree.

---

[9]Invariance based adversarial examples downloaded from `https://github.com/ftramer/Excessive-Invariance`.

Table 13: Mitigating the effect of the input dimension on adversarial vulnerability by correcting the margin enforced by the loss function. Regularization constant $\lambda$ is for $\ell_2$ weight decay. Consistent with (Simon-Gabriel et al., 2019), we use $\epsilon$-FGSM perturbations, the optimal $\ell_\infty$ attack for a linear model. Values in rows with $\sqrt{d} > 28$ are ratios of entry (accuracy or loss) wrt the $\sqrt{d} = 28$ baseline. "Pred." is the predicted increase of the loss $\mathcal{L}$ due to a small $\epsilon$-perturbation using Thm. 4 of (Simon-Gabriel et al., 2019).

| Model | | (Relative) Test Accuracy | | (Relative) Loss | | |
|---|---|---|---|---|---|---|
| $\sqrt{d}$ | $\lambda$ | Clean | $\epsilon = 0.1$ | Clean | $\epsilon = 0.1$ | Pred. |
| 28 | – | $92.4 \pm 0.1\%$ | $53.9 \pm 0.3\%$ | $0.268 \pm 0.001$ | $1.410 \pm 0.004$ | - |
| 56 | – | $1.001 \pm 0.001$ | $0.33 \pm 0.03$ | $1.011 \pm 0.007$ | $2.449 \pm 0.009$ | 2 |
| 56 | 0.01 | $0.999 \pm 0.002$ | $0.98 \pm 0.01$ | $1.010 \pm 0.007$ | $1.01 \pm 0.01$ | - |
| 84 | – | $0.998 \pm 0.002$ | $0.10 \pm 0.09$ | $1.06 \pm 0.01$ | $4.15 \pm 0.02$ | 3 |
| 84 | 0.0225 | $0.996 \pm 0.003$ | $0.96 \pm 0.04$ | $1.05 \pm 0.02$ | $1.06 \pm 0.03$ | - |
| 112 | – | $0.992 \pm 0.004$ | $0.1 \pm 0.2$ | $1.18 \pm 0.03$ | $5.96 \pm 0.02$ | 4 |
| 112 | 0.05 | $0.987 \pm 0.004$ | $1.00 \pm 0.03$ | $1.14 \pm 0.04$ | $1.04 \pm 0.03$ | - |

Table 14: Two-hidden-layer ReLU MLP (see main text for architecture), with and without batch norm (BN), trained for 50 epochs and repeated over five random seeds. Values in rows with $\sqrt{d} > 28$ are ratios wrt the $\sqrt{d} = 28$ baseline (accuracy or loss). There is a considerable increase of the loss, or similarly, a degradation of robustness in terms of accuracy, due to batch norm. The discrepancy for BIM-$\ell_\infty$ with $\epsilon = 0.1$ for $\sqrt{d} = 84$ with batch norm represents a $61 \pm 1\%$ degradation in absolute accuracy compared to the baseline.

| Model | | (Relative) Test Accuracy | | (Relative) Loss | |
|---|---|---|---|---|---|
| $\sqrt{d}$ | BN | Clean | $\epsilon = 0.1$ | Clean | $\epsilon = 0.1$ |
| 28 | ✗ | $97.95 \pm 0.08\%$ | $66.7 \pm 0.9\%$ | $0.0669 \pm 0.0008$ | $1.06 \pm 0.02$ |
| 28 | ✓ | $0.9992 \pm 0.0012$ | $0.34 \pm 0.03$ | $1.06 \pm 0.04$ | $3.18 \pm 0.03$ |
| 56 | ✗ | $1.0025 \pm 0.0009$ | $0.80 \pm 0.02$ | $0.87 \pm 0.02$ | $1.68 \pm 0.03$ |
| 56 | ✓ | $1.0027 \pm 0.0008$ | $0.13 \pm 0.09$ | $0.91 \pm 0.03$ | $5.83 \pm 0.03$ |
| 84 | ✗ | $1.0033 \pm 0.0009$ | $0.71 \pm 0.02$ | $0.86 \pm 0.02$ | $2.15 \pm 0.03$ |
| 84 | ✓ | $1.0033 \pm 0.0010$ | $0.09 \pm 0.08$ | $0.88 \pm 0.02$ | $7.34 \pm 0.02$ |

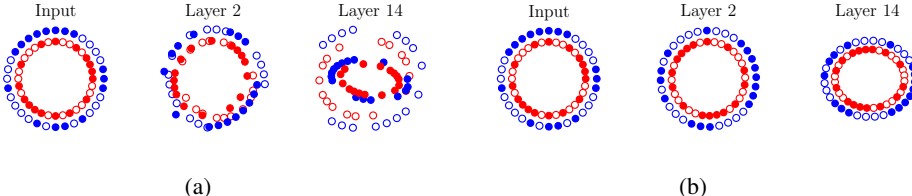

(a)                                           (b)

Figure 8: Two mini-batches from the "Adversarial Spheres" dataset (2D variant), and their representations in a deep linear network at initialization time (a) with batch norm and (b) without batch norm. Mini-batch membership is indicated by marker fill and class membership by colour. Each layer is projected to its two principal components. In (b) we scale both components by a factor of 100, as the dynamic range decreases with depth under default initialization. We observe in (a) that some samples are already overlapping at Layer 2, and classes are mixed at Layer 14.

## G  COMMON CORRUPTION ROBUSTNESS

For VGG8, the mean generalization gaps due to batch norm for noise were: Gaussian—$9.2 \pm 1.9\%$, Impulse—$7.5 \pm 0.8\%$, Shot—$5.6 \pm 1.6\%$, and Speckle—$4.5 \pm 1.6\%$. After the "noise" category the next most damaging corruptions (by difference in accuracy due to batch norm) were: Contrast—$4.4 \pm 1.3\%$, Spatter—$2.4 \pm 0.7\%$, JPEG—$2.0 \pm 0.4\%$, and Pixelate—$1.3 \pm 0.5\%$. Results for the remaining corruptions were a coin toss as to whether batch norm improved or degraded robustness, as the random error was in the same ballpark as the difference being measured.

For VGG13, the batch norm accuracy gap enlarged to $26 - 28\%$ for Gaussian noise at severity levels 3, 4, and 5; and over $17\%$ for Impulse noise at levels 4 and 5. Averaging over all levels, we have gaps for noise variants of: Gaussian—$20.9 \pm 1.4\%$, Impulse—$13.6 \pm 0.6\%$, Shot—$14.1 \pm 1.0\%$, and Speckle—$11.1 \pm 0.8\%$. Robustness to the other corruptions seemed to benefit from the slightly higher clean test accuracy of $1.3 \pm 0.1\%$ for the batch-normalized VGG13. The remaining generalization gaps varied from (negative) $0.2 \pm 1.3\%$ for Zoom blur, to $2.9 \pm 0.6\%$ for Pixelate.

For the WRN, the mean generalization gaps for noise were: Gaussian—$12.1 \pm 2.8\%$, Impulse—$10.7 \pm 2.9\%$, Shot—$8.7 \pm 2.6\%$, and Speckle—$6.9 \pm 2.6\%$. Note that the large uncertainty for these measurements is due to high variance for the model with batch norm, on average $2.3\%$ versus $0.7\%$ for Fixup. JPEG compression was next at $4.6 \pm 0.3\%$.

## H  ADVERSARIAL SPHERES

The "Adversarial Spheres" dataset contains points sampled uniformly from the surfaces of two concentric $n$-dimensional spheres with radii $R = 1$ and $R = 1.3$ respectively, and the classification task is to attribute a given point to the inner or outer sphere. We consider the case $n = 2$, that is, datapoints from two concentric circles. This simple problem poses a challenge to the conventional wisdom regarding batch norm: not only does batch norm harm robustness, it makes training less stable. In Figure 9 we show that, using the same architecture as in (Gilmer et al., 2018), the batch-normalized network is highly sensitive to the learning rate $\eta$. We use SGD instead of Adam to avoid introducing unnecessary complexity, and especially since SGD has been shown to converge to the maximum-margin solution for linearly separable data (Soudry et al., 2018). We use a finite dataset of 500 samples from $\mathcal{N}(0, I)$ projected onto the circles. The unormalized network achieves zero training error for $\eta$ up to 0.1 (not shown), whereas the batch-normalized network is already untrainable at $\eta = 0.01$. To evaluate robustness, we sample 10,000 test points from the same distribution for each class (20k total), and apply noise drawn from $\mathcal{N}(0, 0.005 \times I)$. We evaluate only the models that could be trained to $100\%$ training accuracy with the smaller learning rate of $\eta = 0.001$. The model with batch norm classifies $94.83\%$ of these points correctly, while the unnormalized net obtains $96.06\%$.

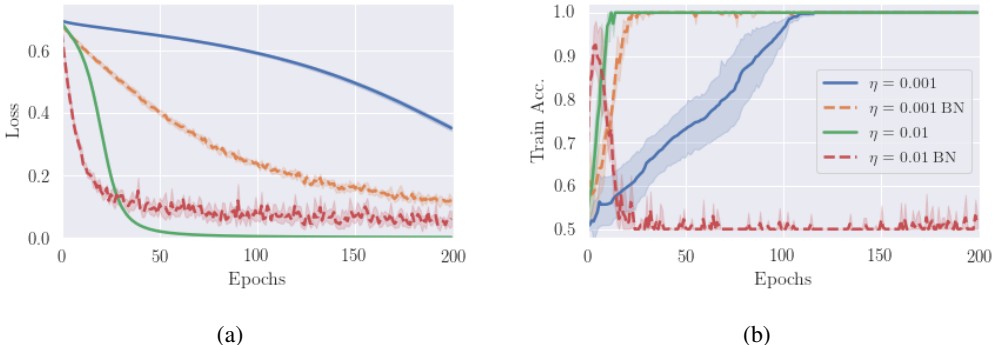

(a) (b)

Figure 9: We train the same two-hidden-layer fully connected network of width 1000 units using ReLU activations and a mini-batch size of 50 on a 2D variant of the "Adversarial Spheres" binary classification problem (Gilmer et al., 2018). Dashed lines denote the model with batch norm. The batch-normalized model fails to train for a learning rate of $\eta = 0.01$, which otherwise converges quickly for the unnormalized equivalent. We repeat the experiment over five random seeds, shaded regions indicate a 95% confidence interval.

$\theta = 88.9$ $\qquad$ $\theta = 75.2$ $\qquad$ $\theta = 50.0$ $\qquad$ $\theta = 18.7$ $\qquad$ $\theta = 0.0$

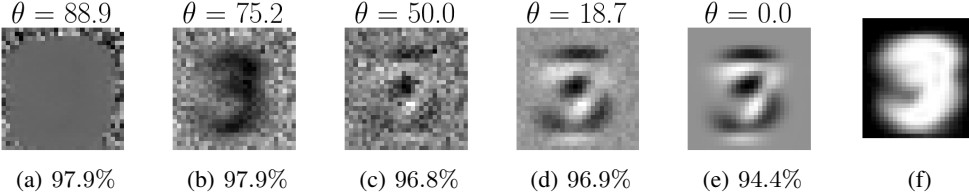

(a) 97.9% $\qquad$ (b) 97.9% $\qquad$ (c) 96.8% $\qquad$ (d) 96.9% $\qquad$ (e) 94.4% $\qquad$ (f)

Figure 10: Weight matrices of various logistic regression models annotated with their tilting angles $\theta$ w.r.t. the nearest centroid classifier (above) and clean test accuracy (below). Plots are (a) BNGD, tested in `eval` mode, (b) BNGD, tested in `train` mode, i.e. without tracking and using the full test set (batch size of 5000), (c) GD baseline with no preprocessing other than division by 255 for pixels in range [0, 1], (d) GD baseline with per-image normalization, (e) the nearest centroid classifier, (f) the standard deviation over the training set. For BN we use $c$=1e-5.

$\theta = 88.9$ $\qquad$ $\theta = 86.8$ $\qquad$ $\theta = 81.8$ $\qquad$ $\theta = 76.1$ $\qquad$ $\theta = 73.4$ $\qquad$ $\theta = 70.3$

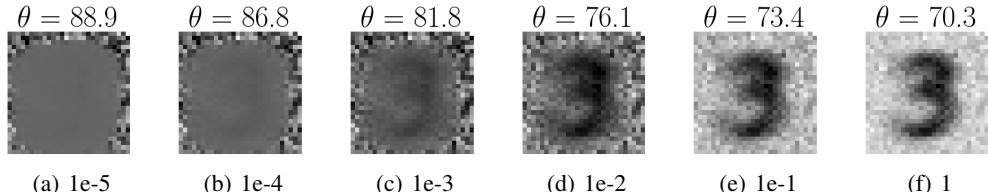

(a) 1e-5 $\qquad$ (b) 1e-4 $\qquad$ (c) 1e-3 $\qquad$ (d) 1e-2 $\qquad$ (e) 1e-1 $\qquad$ (f) 1

Figure 11: By increasing the numerical stability constant $c$ to between 1e-2 and 1e-1, we can achieve the same tilting angle $\theta$ as obtained by not using tracking (10(b)), but which works with arbitrary batch sizes at test time. We test BNGD in `eval` mode (with tracked mean and variance from the training set) and sweep $c$ from the `pytorch` default value of 1e-5 to 1. The weights are then visualized after training.

## I ALTERNATIVE EXPLANATIONS OF THE VULNERABILITY

In a concurrent submission, it is suggested that the main reason for BN induced vulnerability is a mismatch between the mean and variance statistics at training versus test time (Anonymous, 2020). We investigate this hypothesis using the well controlled MNIST 3 vs. 7 testbed from §3.

Figure 10 shows that by eliminating tracking, i.e. going from plot 10(a) to 10(b), the boundary tilting angle $\theta$ w.r.t. the nearest-centroid classifier is reduced by 13.7°. Eliminating BN altogether reduces $\theta$ by an additional 25.2°, and using per-image normalization (rather than per-pixel as in BN) achieves a further reduction of $\theta$ by 31.3°. In terms of FGSM at $\epsilon = 1/10$, model 10(b) (BN, no tracking) achieves 38.7% accuracy whereas the unnormalized baseline 10(c) achieves 66.1%.

The observation regarding tracking is insightful and contributes to a more refined understanding of BN. However, this experiment is a counterexample to the claim that the tracking aspect of BN is the main source of vulnerability, compared to boundary tilting arising from the per-dimension normalization procedure itself. As acknowledged in Anonymous (2020), it is worth re-iterating that using BN without tracked statistics at test-time is restricted to large batch sizes, which may not be practical.

In Figure 11, we show that the reduction in vulnerability achieved by not using tracked statistics at test time can be achieved by increasing the numerical stability constant $c$. Here, the $c$ value can be used to interpolate between the configurations achieved when tracking is used (shown in Figure 10(a)), or not used (shown in Figure 10(b)) for a fixed $c$.

There remains a distinct qualitative difference between the weight matrices visualized in Figures [10(b) ($c$=1e-5, no tracking) – 11(f) ($c$=1, with tracking)], which both resemble the hand-drawn digit "3", compared to the case without BN in Figures [10(c)–(e)] which contain both a "3" as dark pixels and a "7" as bright pixels. This is a consequence of the BN procedure not being class-aware, i.e., all pixels of a digit "7" are normalized by the variance of both class "3" and class "7", which more closely resembles a "3" for this dataset as shown in Figure 10(f).

## J    ADVERSARIAL EXAMPLES

To alleviate possible concerns that *gradient masking* effects (Papernot et al., 2017; Athalye et al., 2018) may explain the apparent difference in robustness between batch-normalized and unnormalized networks, we provide qualitative evidence that BN degrades the visual appearance of adversarial examples for strong unbounded attacks. In other words, we show that these unbounded adversarial examples remain "adversarial" for batch-normalized networks, while semantically relevant features are introduced for inputs classified with high confidence by the baselines, such that a human oracle would agree with the predictions. If the gradients were obfuscated in any way, we would be unable to meaningfully increase the accuracy or prediction confidence of the models beyond random chance, and the examples that we initialize from white noise would remain noisy. Note: this is not the sole source of evidence we provide that gradient masking is not a factor, the common corruption benchmarks and unbounded attacks reaching $100\%$ success also support this.

### J.1    MNIST

For the MNIST experiments, we train a LetNet-5 CNN variant from the `AdverTorch` library (Ding et al., 2019). All models (batch-normalized or not) are trained for 90 epochs of SGD with a constant learning rate of 0.01, and a mini-batch size of 50. We compare three variants of this model:

1. **L2**: Uses an $L_2$ weight decay regularization constant $\lambda = 0.01$.
2. **BN**: The two convolution layers of the LetNet-5 are batch-normalized, with default settings ($\gamma$ and $\beta$ enabled, numerical stability constant set to 1e-5, momentum of 0.1).
3. **Per-img**: Subsumes "L2" and additionally uses per-image normalization, i.e., we subtract the global mean of each image to correct the average intensity, then divide by the standard deviation of all features in the image.[10]

The models obtain similar clean test accuracy: "L2"—$98.0\%$, "BN"—$99.2\%$, "Per-img"—$98.4\%$.

To construct adversarial examples, two unbounded attack schemes are used: "fooling images" (Nguyen et al., 2015) and the Carlini & Wagner (2017) (CWL2) method. The fooling images procedure starts with white noise, then minimizes the cross entropy loss wrt a target class until it is predicted with high confidence. "Confidence" here means the softmax margin, i.e., the difference between the max softmax output and the second from max output. Formally, we sample a starting point $x^0 \sim \mathcal{N}(0, 0.1) \in \mathbb{R}^{28 \times 28}$, then update $x$ according to $x^n = x^{n-1} + \delta$, using $n = 1$ to 100 iterations of PGD-$\ell_2$ w.r.t. each class label. For "L2" and "BN", we clip pixel values to $[0, 1]$, and for "Per-img" to $[-1, 1]$. We use a PGD-$\ell_2$ step size of 0.2.

Figure 12 shows the fooling images, their predicted class labels, and prediction confidence for the three configurations. The procedure bears similarity to "Activation Maximization" techniques for interpreting neural networks, but this usually requires post-hoc regularization of the input images (Simonyan et al., 2014). This was not required here, owing to the competitive robustness of models 12(a) and 12(b), which makes them somewhat interpretable by default. The batch-normalized model classifies images that only faintly resemble natural digits with full confidence, while those of the baselines contain more clearly class-relevant features. In particular, the "Per-img" case in Figure 12(b) makes no fully saturated confidence predictions, and its fooling images resemble matched filters (Turin, 1960) w.r.t. task relevant edges.

The Carlini & Wagner (2017) (CWL2) Adam optimizer-based attack seeks to find the smallest perturbation in terms of $\ell_2$ distance which achieves either an arbitrary, or targeted, misclassification. We use the arbitrary misclassification objective, confidence parameter $k = 4$ to encourage reasonably high confidence predictions, a "learning rate" of 1e-2, nine (9) binary search steps, and an initial value for the constant $c$ of 1e-3, which balances the minimization of the $\ell_2$-norm of the perturbation $\delta$, and prediction confidence.[11] We allow up to 10,000 attack iterations, and show the resulting examples in Figure 13. For the same confidence threshold, the mean $\ell_2$ norm of the perturbations required

---

[10]This contrasts sharply with BN which computes the same statistics but over samples, and then normalizes each dimension.

[11]Note that $c$ here is a meta-parameter of the CWL2 attack and is not to be confused with the BN numerical stability constant.

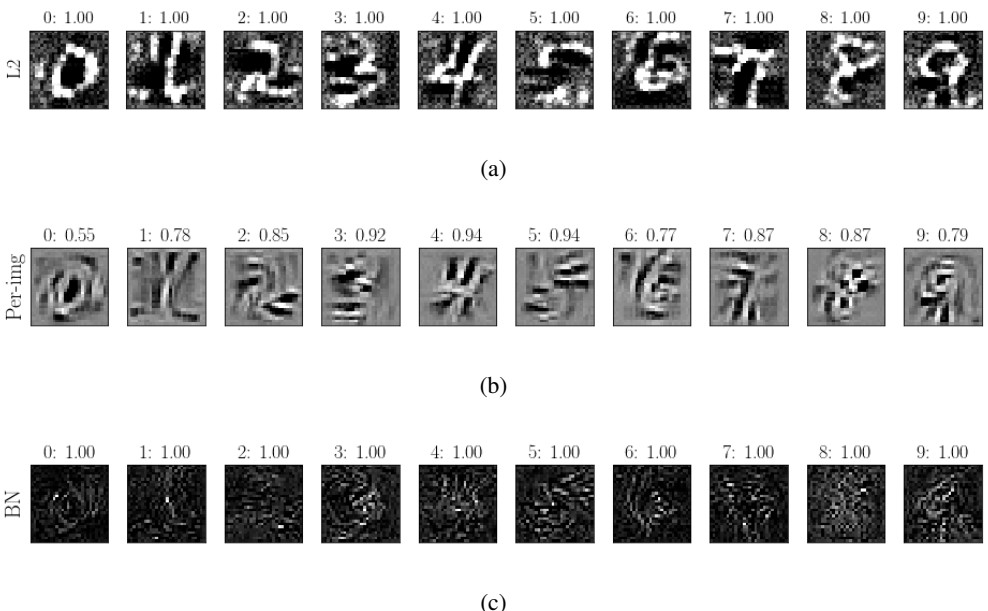

Figure 12: Fooling images crafted on three variants of LeNet-5 Model. Case (a) is the $L_2$ weight decay baseline, case (b) subsumes (a) and includes *per-image* normalization. In case (c), the convolution layers of the LeNet are batch-normalized and no other explicit regularization is used. See text for detailed training and evaluation meta-parameters.

to change the predictions of the batch-normalized model was $\|\delta\|_2 = 0.95$, while $\|\delta\|_2 = 2.89$ was required for the unnormalized model. This is a three fold improvement in terms of the relevant performance metric for this attack, which always obtains $100\%$ success by virtue of being unbounded.

## J.2 SVHN

For SVHN, we visualize the fooling images in Figure 14, and all combinations of source to targeted misclassification crafted using a basic iterative gradient method (BIM-$\ell_2$) for a random sample of each class in Figure 15.

We use a simple CNN architecture described in Table 15. The model is trained for 50 epochs of SGD on the SVHN dataset (excluding the "extra" split) with a constant learning rate of 0.01, and a mini-batch size of 128 in both cases. Since SVHN is imbalanced, the loss is weighted by the inverse frequency of each class. As in §J.1, for the no-BN baseline we use an $L_2$ weight decay regularization constant $\lambda = 0.01$, and per-image preprocessing.

Although both models obtain similar a test accuracy of about $90\%$, the fooling images for the unnormalized model contain task relevant edges, like those of Figure 12(b), whereas the batch-normalized model yields difficult to interpret images. Similarly, the examples of grid 15(a) resemble the target class in many cases, while those of 15(b) for the batch-normalized model have a subtle textured effect, and preserve the semantic cues of the source image.

## K ON THE INITIAL LEARNING RATE

It is known that using a high initial learning rate, which is annealed during training, often achieves higher test accuracy than training with a smaller fixed learning rate, even though the latter approach optimizes the training loss more quickly. To explain this phenomenon, Li et al. (2019) introduce the concept of *learning order*, which is a training time property that was found to be predictive of generalization ability, in contrast with *post-training* notions of model complexity that are typically believed to govern generalization. Learning order refers to different examples being learned at

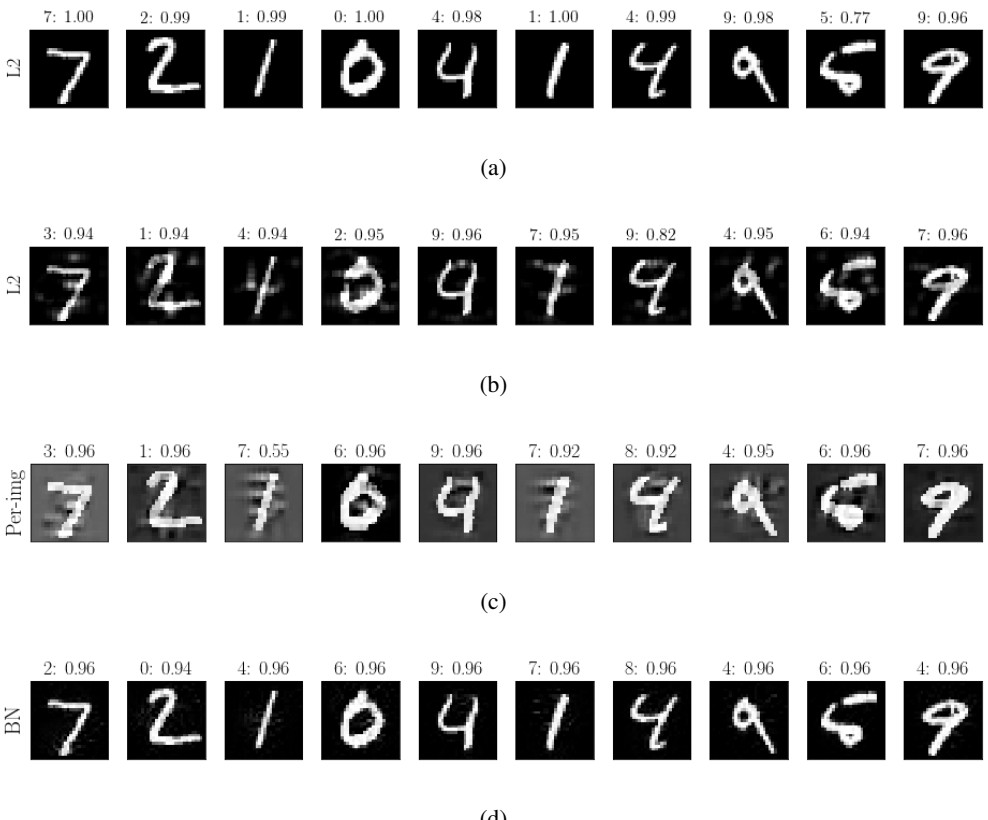

Figure 13: Candidate adversarial examples crafted with the Carlini & Wagner (2017) (CWL2) attack on originally correctly classified images (a). For CWL2 we set the confidence parameter $k = 4$, and use the "arbitrary misclassification" objective, i.e., the adversary only needs to flip the label, not achieve any particular label. Plot (b) shows the perturbed images ($x_{\text{adv}} = x + \delta$) for the baseline $L_2$ regularized model ($\|\delta\|_2 = 2.89$), while plot (d) shows those of the batch-normalized model. The "adversarial examples" shown in (b) contain semantic features corresponding to the predicted class. An image which was originally correctly classified as the digit "1" is now classified as "7", but the top of a "7" has been added, as well as the commonly used "dash" through the center. Several instances of "4" are turned into "9", as the top of the loop of the "4" is completed. For the batch-normalized model, whose adversarial examples are shown in (d), the examples are classified with the same or higher confidence, yet the perturbations are not perceptible and their $\ell_2$-norms are significantly lower ($\|\delta\|_2 = 0.95$).

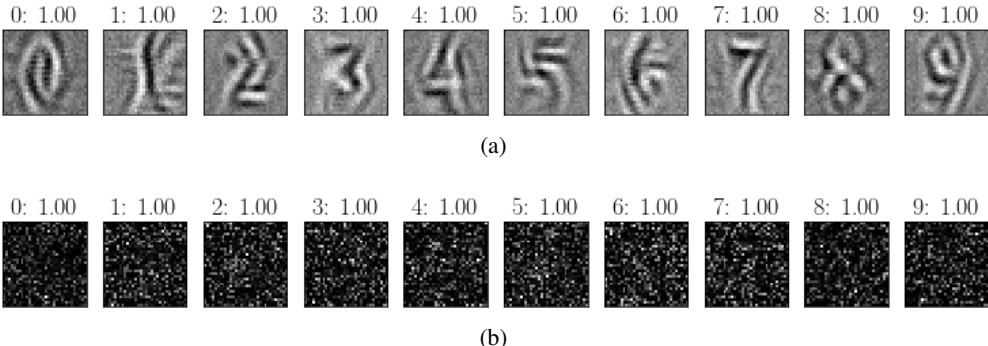

Figure 14: Fooling images crafted on a simple CNN (a), and the same CNN with two batch-normalized convolution layers (b). See text for detailed training and evaluation meta-parameters.

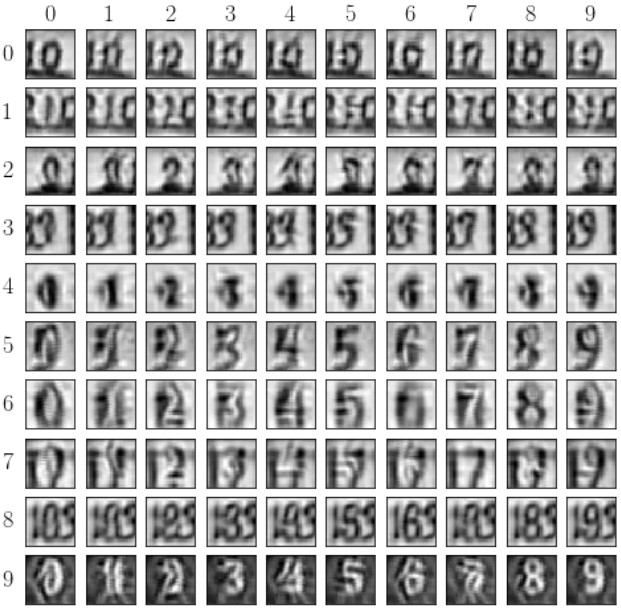

(a)

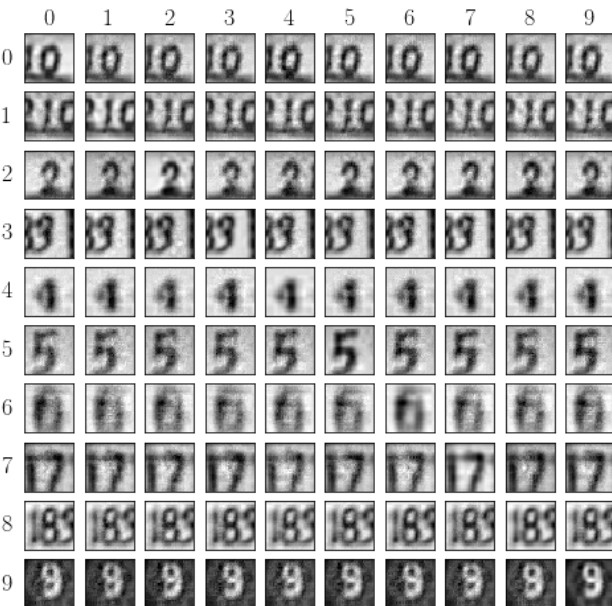

(b)

Figure 15: All source–target transformations obtained by iterating BIM-$\ell_2$ until the baseline (a) predicts the target class (enumerated over the columns) with a mean margin of $90\%$. The source class is enumerated along the rows such that the natural images are on the diagonal. (b) is batch-normalized.

Table 15: CNN architecture adapted from the `CleverHans` library (Papernot et al., 2018) for the SVHN fooling image experiment. We denote the convolution kernel height, width, number of input and output channels w.r.t. each layer, and stride as: $h, w, c_{in}, c_{out}, s$, respectively. For the BN variant, we apply BN with default meta-parameters after `Conv1` and `Conv2`.

| Layer | $h$ | $w$ | $c_{in}$ | $c_{out}$ | $s$ | params |
|---|---|---|---|---|---|---|
| Conv1 | 8 | 8 | 1 | 32 | 2 | 2.0k |
| Conv2 | 6 | 6 | 32 | 64 | 2 | 73.8k |
| Conv3 | 5 | 5 | 64 | 64 | 1 | 102.4k |
| Fc | 1 | 1 | 256 | 10 | 1 | 2.6k |
| Total | – | – | – | – | – | **180.9k** |

different times, according to their intrinsic noise level and complexity. Li et al. find that using a high initial learning rate prevents the model from fitting high complexity patterns too early, at the expense of easier to fit patterns. When the learning rate is annealed, the model effectively moves on to the next stage of a "curriculum" in which it can learn more challenging patterns.

To examine the interplay between the learning rate and robustness, we conduct a similar experiment on CIFAR-10 as in Li et al. (2019) with "small" and "large" initial learning rate ("lr"). We evaluate several measures of robustness every 10 epochs to better characterize the regimes in which the vulnerability occurs, and its relationship with the annealing of the learning rate. A shallow VGG variant (VGG8) is trained over 150 epochs using two different initial learning rates (details in caption of Figure 16), and the process is repeated over three random seeds for BN vs. no-BN. We use standard meta-parameters, SGD with momentum 0.9, a mini-batch size of 128, weight decay 5e-4, and standard data augmentation. We use "Un" to denote the unnormalized model, and "BN" for batch-normalized. The final test accuracies are: "Un, small lr": $88.7 \pm 0.1\%$, "Un, large lr": $90.4 \pm 0.1\%$, (diff. $1.7 \pm 0.1\%$), "BN, small lr": $90.3 \pm 0.1\%$, "BN, large lr": $91.3 \pm 0.1$, (diff. $1.0 \pm 0.1\%$).

To evaluate robustness, we evaluate accuracy on several variants of the test set: "Clean"—the original test set, "AWGN"—with additive white Gaussian noise ($\sigma^2 = 1/16$), "FSGM"—with perturbation crafted by one-step fast gradient sign method $\ell_\infty$, and "PGD"—a 40-step PGD-$\ell_\infty$ perturbation, with a step size of $\epsilon/30$. We use $\epsilon = 8/255$ as the $\ell_\infty$ norm budget for both FGSM and PGD perturbations, and clip the pixel values to $[\pm 2]$.

For the "high lr" case shown in Figure 16(a), the unnormalized model achieves higher robustness than the batch-normalized model after around 70–80 epochs, shortly after the learning rate is first annealed. Meanwhile, Figure 16(b) shows that in the "small lr" case, the unnormalized model achieves higher robustness on all tests compared to its batch-normalized equivalent, at all epochs. The "high lr" result of Figure 16(a) begets the question: if we are willing to sacrifice clean test accuracy and *do not* anneal the large learning rate, can we mitigate the BN vulnerability? We investigate this in a subsequent experiment depicted in Figure 17, which indeed shows that there appears to be little difference in robustness for this case. We conclude with a few remarks:

1. The "large constant learning rate" is not a particularly practical setting, as $10\%$ absolute clean test accuracy has been left on the table.

2. We interpret the "large constant learning rate" result not as "the vulnerability does not occur", but rather, the baseline fails to reach its potential. Indeed, from Figure 16(b) and Table 2, training the same architecture with a small learning rate yields $52.9 \pm 0.6\%$ for the same PGD-$\ell_\infty$ perturbation, while the BN variant achieves less than $40\%$ accuracy at all times (this can be seen by inspection of Figures 16 and 17).

3. The accuracy of the unnormalized model on *all* test sets increases almost monotonically with prolonged training, and appears to be still increasing at the conclusion of the 150 epochs. For fairness, we did not investigate further training to see how long this trend continues. Conversely, the batch-normalized model converges quickly then plateaus. At this point its accuracy tends to oscillate, or even *decline monotonically* (particularly for PGD in Figure 16(b)), thus careful early stopping is much more relevant to BN.

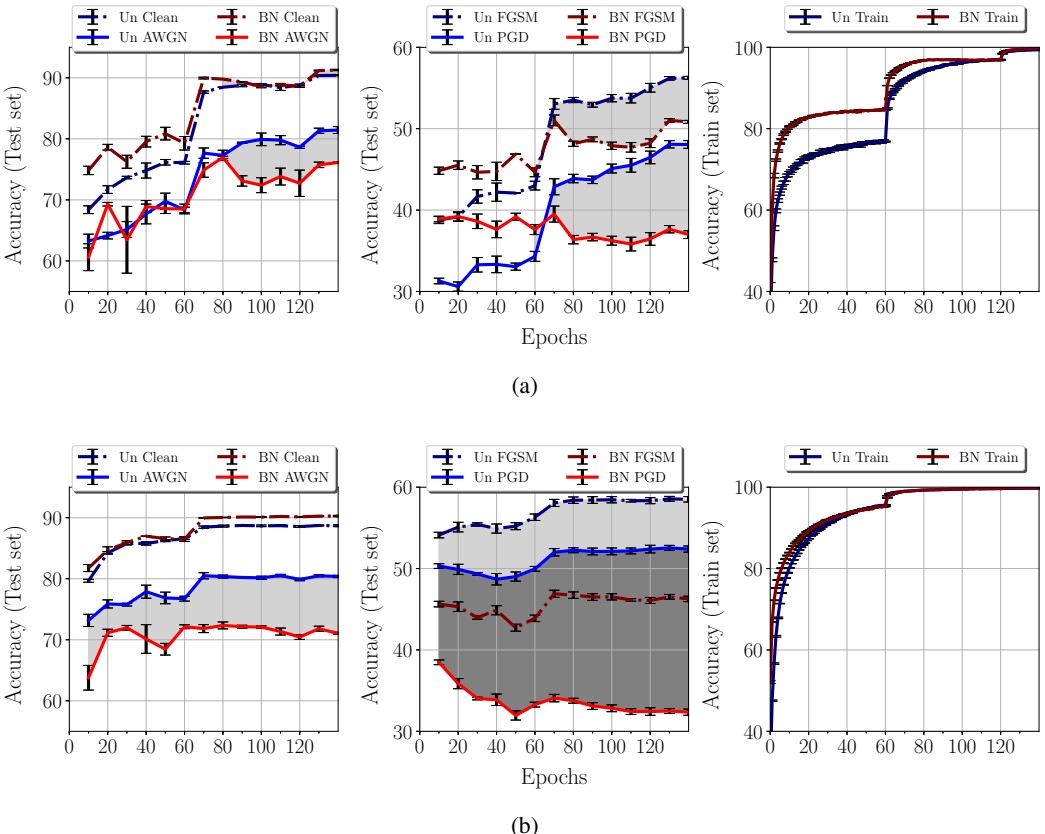

Figure 16: We train a VGG model on CIFAR-10 for 150 epochs using a "large" (a) and "small" (b) initial learning rate (0.1, and 0.01 respectively), which is dropped by a factor of ten at epochs 60 and 120 (best seen in the training curves of the right column). Every ten epochs, we evaluate the accuracy on several variants of the test set: "Clean"—the original test set, "AWGN"—with additive white Gaussian noise, "FSGM"—with fast gradient sign method one-step $\ell_\infty$ perturbation, and "PGD"—a 40-step PGD-$\ell_\infty$ perturbation. See text for additional meta-parameters. The light grey shading indicates where the accuracy of the unnormalized model exceeds that of the batch-normalized equivalent for a given test. Note: the series "Un PGD" achieves higher accuracy than "BN FGSM" in (b), so we use two different shades to distinguish the overlapping FGSM and PGD areas. As expected, the FGSM curve lies above the PGD curve corresponding to the same model at all points. Error bars indicate the standard error over three random seeds.

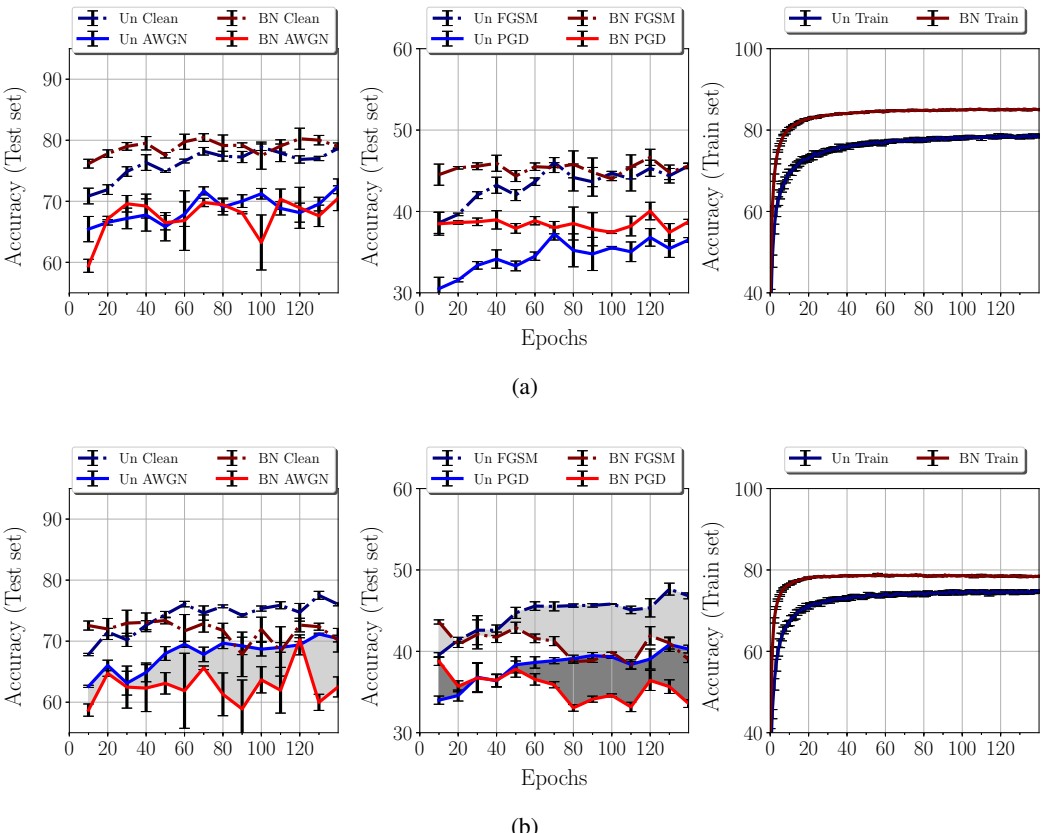

Figure 17: To isolate the effect of annealing the learning rate for CIFAR-10, we repeat the experiment of Figure 16, only this time using a large learning rate (0.1) that is *fixed* during training. In (a), we have our first negative result: the accuracy of the BN variant is either compatible, or slightly higher than that of the unnormalized variant. We discuss the implications of this result in the text. In (b), we double the weight decay penalty to 1e-3. For the sake of completeness, we do this for both BN and no-BN cases, even though weight decay seems to have a different mechanism when combined with BN, to increase the effective learning rate (van Laarhoven, 2017; Zhang et al., 2019a). Thus, increasing the weight decay leads to instability in the BN case, as the learning rate is already large. The relevant comparison is therefore between the unnormalized curves of (b), and the BN curves of (a), in which the higher weight decay unnormalized model is competitive with, and slightly outperforms by epoch 130, the better performing batch-normalized model in terms of robustness. Error bars indicate the standard error over three random seeds.

