# OpenReview forum: "Batch Normalization is a Cause of Adversarial Vulnerability"
_ICLR.cc/2020/Conference — Reject_

### Official Review · AnonReviewer3 · 2019-10-11
**Official Blind Review #3**

**Rating:** 3

**Review:**

This paper identifies an important weakness of batch normalization: it increases adversarial vulnerability. It is very well written and the claims are theoretically sound. In the experiments, the authors demonstrated a significant difference in robustness between networks with or without batch normalization layers, in varies settings against both random input noise and adversarial noise. This weakness of batch norm was explained due to the "decision boundary tilting" effect caused by the normalization. Overall, this paper has done solid work to reveal an interesting phenomenon. If it is true, this finding will impact almost all DNN models.

My concern is that this phenomenon is just another effect of "gradient masking " (as pointed out by Athalye, et al.). Batch norm is a well-known technique to avoid overfitting, without batch norm the network can be easily trained to be saturated with almost zero gradients, demonstrating a false signal of "robustness" to noise. The random noise and real-world corruption experiments are definitely helpful to clear this doubt, but only partially. My concern remains because of two obvious signs of  gradient masking:
1. The accuracy on PGD-li (epsilon=0.031) attacks are suspiciously too high (20% - 40% Table 3/4). For this level of attack, the acc should be nearly zero. This is likely caused by the gradient masking effect, considering the cifar-10 networks were trained for longer time with larger learning rate (150 epochs, fixed lr 0.01). Training on MNIST is much easier to get zero gradients.
2. The weight decay discussion is not helpful at all, on the contrary, it confirms my concern on the gradient masking effect. In Table 8, the robustness was increased ~40% by just using large weight decay. This is not the "real robustness", and can be easily evaded by adaptive attack (see Athalye's paper).
With the above two concerns in mind, I doubt the phenomenon revealed in this paper is just "one can easily train a saturated model without batch norm" or equivalently "it's hard to train a saturated model with batch norm". It is hard to say if this is a bad thing for batch norm.

I am quite surprised that the authors ignore this completely. Here are a few things that can be done to rule out the possibility of gradient masking. The masked gradient can be identified by: 1) One-step attacks perform better than iterative attacks; 2) Unbounded attacks do not reach 100% success., etc (see Section 3.1 of Athalye's paper).
1. Including FGSM in the experiments and show the same trends as PGD-li.
2. Show two networks have similar gradient norms.
3. Apply cw-l2 attack, and show batch norm has forced large perturbation.

Two other suggestions:
1. Summarize the different angles/steps taken to verify the phenomenon, somewhere before the experiments.
2. Cannot see why the input dimension discussion contribute to explanations of the batch norm weakness.

============
My rating stays the same after rebuttal.

My original concerns are like the other reviewers: why BN, not other techniques such as structure of DNNs MLP vs CNN vs ResNet, activation functions, weight decay, learning rates, softmax etc. My initial suspect was that it is caused by gradient masking likely caused by the l2 weight regularization, so asked the authors to look at the gradient norms and run some testes to rule this out. Yes, the weight norm is directly related to the Lipschitz continuity of the function represented by the network, but it often becomes more complicated on complex nonlinear neural networks.

According to the new experiment results, the vulnerability is indeed not an effect of gradient masking, thanks for the clarification. However, the new results also indicate that the finding is susceptible to both weight decay and learning rate: in Figure 16 (a): "Un PGD" < "BN PGD" before learning rate decay, andFigure 17 (a) vs (b), doubling the weight decay penalty to 1e-3 also increases the vulnerability of BN. Overall, I believe the phenomenon exists, but the reasons behind requires more explanations, at least not just the batch norm.

**Experience Assessment:**

I have published one or two papers in this area.

**Review Assessment: Checking Correctness Of Derivations And Theory:**

I assessed the sensibility of the derivations and theory.

**Review Assessment: Checking Correctness Of Experiments:**

I carefully checked the experiments.

**Review Assessment: Thoroughness In Paper Reading:**

I read the paper thoroughly.

---

> ### Author Response · Authors · 2019-11-05
> **Clarification about gradient masking**
>
> We thank the reviewer for recognising the positive aspects of this work, and for stating that they believe the work to be theoretically sound. We agree that it's imperative to ensure results are not tainted by gradient masking; we have taken care to ensure this did not play a role here and wish to alleviate this natural concern.
>
> --
>
> Re: 1. “The accuracy on PGD-li (epsilon=0.031) are suspiciously too high (20% - 40% in Tables 3, 4), the accuracy should be nearly zero”.
>
> For the experiments of Section 4, the input was normalized to zero mean and unit variance using per-channel statistics computed per dataset. Thus, epsilon=0.03 represents about 4/255 rather than 8/255 on the [0, 1] scale. In Figure 5, epsilon is increased until zero accuracy is reached for a vanilla ResNet, and accuracy is indeed reduced to zero by epsilon=6 for all models under standard training. The difference in accuracy at epsilon 4/255 compared to Tables 3/4 is explained by the difference in architecture, WideResNet and VGG. In particular, VGG is more robust than the residual networks we tested under standard training.
>
> --
>
> Re: 2. “The weight decay discussion is not helpful at all, on the contrary, it confirms my concern on the gradient masking effect. In Table 8, the robustness was increased ~40% by just using large weight decay. This is not the "real robustness", and can be easily evaded by adaptive attack (see Athalye's paper).
>
> 1. The claim you mention (40% increase in robustness) was evaluated using an adaptive method.
>
> 2. We interpret this comment as an affirmation that our findings are non-obvious, rather than as a limitation.
>
> 3. We would really appreciate if you can clarify why you believe the weight decay discussion is not helpful, and the robustness “not real”. There are strong theoretical connections between penalizing the parameter norms, robust optimization, and decision boundary tilting, see e.g., Xu & Manor, (JMLR 2009), Tanay & Griffin, (2016). We also showed that this is essential for mitigating an increase in vulnerability as the input dimension increases. L2 weight decay minimizes an upper bound on the Frobenius norm of the linear operators in the network, thus bounding the Lipschitz constant of the network. The Lipschitz constant taken together with the mean prediction margin is well known to govern adversarial perturbation robustness (Tsuzuku et al., in NeurIPS 2018).
>
> 4. Can you recommend a specific attack you would like us to evaluate against that would increase your confidence in our results? We are familiar with the work of Athalye et al and do not believe that the attack evaluation is in any way related to the "defense", all attacks are unseen to all models as of training time. Furthermore, a concurrent submission and Reviewer 2 appear to have successfully reproduced our main result.
>
> --
>
> Re: "One-step attacks perform better than iterative attacks".
>
> Are you referring to an instance of this rule being violated in our work (if you believe you saw this, it would help us if you could point to a specific instance), or are you suggesting that we show results for one step attacks alongside those of iterative attacks? It's generally agreed that one step attacks aren't as meaningful for deep/nonlinear models, which is why we only use FGSM for linear models and PGD for deep models. Nonetheless, we have added this result to a new section of the Appendix titled “On the Initial Learning Rate” where we plot the test accuracy under various perturbations vs training epochs. The FGSM curves lie above the 40-step PGD curve for a given model in all cases.
>
> Also, Appendix C deals with unbounded attacks which reach 100% success in all cases.
>
> --
>
> Re: “Apply cw-l2 attack, and show batch norm has forced large perturbation.”
>
> Thank you for this suggestion. We have added adversarial examples crafted by the CWL2 method in Figure 13 of an Appendix I titled “Adversarial Examples”. The L2 distortion required to achieve a fixed misclassification confidence threshold is 0.95 in the case of batch norm, and 2.89 for the baseline, which represents a three fold improvement on the relevant performance metric for this attack.
>
> The white-box procedures would not work without access to a clean gradient signal, yet all reach 100% success, or confidence, in all cases.
>
> Please let us know if these clarifications address your concerns. We agree that our work would be far less impactful if our result could be reduced to gradient masking. We hope that the additional experiments and analysis we have performed lays this concern to rest.

---

> > ### Author Response · Authors · 2019-11-13
> > **References**
> >
> > References
> >
> > Huan Xu, Constantine Caramanis, and Shie Mannor. Robustness and Regularization of Support VectorMachines. In Journal of Machine Learning Research, 10:1485–1510, 2009
> >
> > Anh Nguyen, Jason Yosinki, Jeff Clune. Deep Neural Networks are Easily Fooled: High Confidence Predictions for Unrecognizable Images. In IEEE Conference on Computer Vision and Pattern Recognition, pp. 427-436, 2015.
> >
> > Yusuke Tsuzuku, Issei Sato, and Masashi Sugiyama. Lipschitz-Margin Training: Scalable Certification of Perturbation Invariance for Deep Neural Networks. In Advances in Neural Information Processing Systems 31, pp. 6541-6650, 2018.

---

### Official Review · AnonReviewer1 · 2019-10-14
**Official Blind Review #1**

**Rating:** 1

**Review:**

Summary:
In this empirical study, the authors identify that batch normalization -- a common technique for accelerating training -- leads to brittle representations that exhibit a lack of robustness and are more susceptible to adversarial attacks. The authors demonstrate their results on SVHN, CIFAR-10, CIFAR-100 CIFAR-10.1 using a variety of network architectures including VGG, BagNet, WideResNet, AlexNet, etc.

Major Concerns:

1. As presented, the experiments are not convincing.

I do not know how much of the changes in adversarial vulnerability are due to batch normalization as opposed to other facets of the training procedure that may have changed in their BN vs no-BN experiments.

For instance, batch normalization usually accommodates higher learning rates and it is not clear if the authors adjusted the initial learning rate, learning rate schedule or training schedule accordingly. If so, it would be important to run a set of experiments with these parameters fixed as per the baseline no-BN models.

That said, even if the authors did run these experiments, it is still not clear if the cause of adversarial vulnerability is due to BN. Consider that what is truly important in model training is not the learning rate (i.e. step size), but rather the magnitude of the changes in each weight (or the ratio of weight change to the weight). By swapping in batch normalization, the authors may just be altering the norm of the weight change in the (re-parameterized) weights. In this scenario, the gains of removing batch normalization could just as well be explained by the effective change in the learning rate, and not about batch normalization itself, c.f.
  Towards Explaining the Regularization Effect of Initial Large Learning Rate in Training Neural Networks
  Yuanzhi Li, Colin Wei, Tengyu Ma
  https://arxiv.org/abs/1907.04595
If the differences in the adversarial vulnerability could be ascribed to effective changes in gradient updates, then this would change the interpretation of these results notably.

2. The underlying hypothesis is specious.

I have several reservations about the underlying hypothesis that requires stronger evidence to overcome. In particular, I have reservations in believing that BN itself is a cause of adversarial vulnerability because BN is just a factorization of a network's weights. That is, there is nothing "special" nor unique about BN-networks; instead, the BN factorization merely permits accelerated training efficiency.

Consider the fact that a BN model may be re-expressed by merely folding in the parameters (i.e. applying the matrix multiplications) into the MLP weights or CNN filters. Thus, the numerical function approximated by the BN and the "folded" non-BN model is identical. What would it mean to say that the BN is "causing" adversarial vulnerability in the BN model given that both the BN and non-BN model perform the identical function?

Another way to say this is to pretend we train a non-BN MLP or CNN model. After training the model, we could apply a BN factorization of the weights. Thus, the non-BN model may be factorized into a BN model. If the resulting BN model were adversarial vulnerable (which I suspect is the case), it would seem very hard to believe that BN was the cause of the vulnerability given it was a post-hoc factorization of the weights.

That said, I could definitely imagine that the training procedure itself could lead to adversarial vulnerability (e.g. citation above) and by employing a BN factorization, one may be encouraged to use a training procedure which leads to increased vulnerability. I would encourage the authors to consider this line of attack and thus, re-orient their analysis and discussion accordingly.

3. The title is poorly worded.

Not withstanding the point above, adversarial vulnerability predates BN. Likewise, non-BN models exhibit adversarial vulnerability. Thus, this title is not a great reflection of the findings of the paper. I would strongly suggest replacing "is a cause" with "increases" or "exacerbates".

**Experience Assessment:**

I have published one or two papers in this area.

**Review Assessment: Checking Correctness Of Derivations And Theory:**

I assessed the sensibility of the derivations and theory.

**Review Assessment: Checking Correctness Of Experiments:**

I carefully checked the experiments.

**Review Assessment: Thoroughness In Paper Reading:**

I read the paper at least twice and used my best judgement in assessing the paper.

---

> ### Author Response · Authors · 2019-11-13
> **Response (part 1) - misc clarifications, new experiments on initial learning rate and folding BN statistics into weights**
>
> We thank the reviewer for their frank comments and pointing us to the work on explaining the effect of using a large initial learning rate. Responding to this review has helped us improve the work, as well as our own understanding. We aim to satisfy the reviewer’s concerns with a new section titled “On The Initial Learning Rate” in which we evaluate robustness on CIFAR-10 during training under various initial learning rates and schedules. In-line responses/clarifications to the review follow:
>
> --
>
> "The authors demonstrate their results on SVHN, CIFAR-10, CIFAR-100 CIFAR-10.1"
>
> To clarify, we evaluated on CIFAR-{10, 10.1, 10-C}, but not on CIFAR-100. We also evaluated pre-trained models on ImageNet, as well as the Adversarial Spheres dataset (Gilmer et al., ICLR Workshop 2018) (albeit in an Appendix).
>
> --
>
> "I do not know how much of the changes in adversarial vulnerability are due to batch normalization as opposed to other facets of the training procedure that may have changed in their BN vs no-BN experiments."
>
> No aspect of the training procedure was changed for BN vs no-BN, unless explicitly stated otherwise, e.g., to allow BN a larger initial learning rate, see next bullet.
>
> --
>
> “batch normalization usually accommodates higher learning rates and it is not clear if the authors adjusted the initial learning rate”
>
> Quoting from the paper at the time of submission:
> (In ref Table 4): “It has been suggested that one of the benefits of BN is that it facilitates training with a larger learning rate (Ioffe & Szegedy, 2015; Bjorck et al., 2018). We test this from a robustness perspective in an experiment summarized in Table 4, where the initial learning rate is increased to 0.1 when BN is used.”
>
> To prevent this from being missed and improve the clarity of our work, we can summarize the scenarios considered at the beginning of Section 4, e.g., “Case 1) both models get small LR”, “Case 2) BN gets high initial LR”. If the reviewer previously read this and still believes it to be unclear, please let us know.
>
> Note that we also provide a counterexample to the conventional wisdom that BN allows a higher learning rate in Appendix G on the Adversarial Spheres dataset.
>
> --
>
> "By swapping in batch normalization, the authors may just be altering the norm of the weight change in the (re-parameterized) weights. In this scenario, the gains of removing batch normalization could just as well be explained by the effective change in the learning rate, and not about batch normalization itself, c.f. (Li et al., 2019)."
>
> As indicated by the two cases (on the synthetic and MNIST dataset) where we compute the boundary tilting angle, BN affects not only the norm (invariance) of the weights, but also the angle. Consider the first thing that happens when we feed forward a batch of inputs after random initialization: the weights are rescaled by the inverse of the standard deviation (and numerical stability const) along each dimension. Thus, the weights corresponding to low variance features increase in value, while those corresponding to high variance features shrink, and the resulting batch-normalized weight vectors can be nearly orthogonal to those without BN. Thus, the angle is a critical difference between batch-normalized and unnormalized weights. Although we motivated and explicitly characterized this for linear models, Section 6 of Labatie, (ICML 2019) shows that deep batch-normalized networks accordingly suffer from increased sensitivity w.r.t. the input as a result, and similarly remark: "[under BN] directions of high signal variance are dampened, while directions of low signal variance are amplified. This preferential exploration of low signal directions naturally deteriorates the signal-to-noise ratio and amplifies $\chi^l$". Where $\chi^l$ is defined as the normalized sensitivity from layer 0 to $l$, such that $\chi^l > 1$ degrades the signal-to-noise ratio.
>
> We found Li et al., 2019 and the learning order concept interesting. We agree that in the context of this work, it is imperative to consider the case where a higher initial learning rate is used with BN given that it does usually facilitate this. Please see Table 4, and the new section “On The Initial Learning Rate” regarding this point which shows that over the course of 150 epochs of training using several learning rate schedules, BN obtains at most 40% accuracy to 40-step PGD while the baseline exceeds 50% accuracy on the same.

---

> > ### Author Response · Authors · 2019-11-13
> > **Response (part 2)**
> >
> > “there is nothing "special" nor unique about BN-networks; instead, the BN factorization merely permits accelerated training efficiency. Consider the fact that a BN model may be re-expressed by merely folding in the parameters (i.e. applying the matrix multiplications) into the MLP weights or CNN filters. Thus, the numerical function approximated by the BN and the "folded" non-BN model is identical.”
> >
> > As outlined by existing theoretical work, training with BN is unique in several respects. To start, it imposes a hard upper limit on maximum trainable depth that is solely a function of the mini-batch size due to gradient explosion (Thms 3.9 & 3.10, Yang et al., 2019), and leads to exploding sensitivity wrt the input (S6 & Fig 4, Labatie et al., 2019).
> >
> > Note: in response to a concurrent submission which suggests that the use of tracked statistics are the source of adversarial vulnerability, we’ve added an Appendix H “Alternative Explanations of the Vulnerability” in which we isolate the effect of folding the BN statistics into the weights post-training on MNIST. The weights learned under BN, either with or without plugging in the tracked statistics at test time, differ substantially both qualitatively, and quantitatively in terms of boundary tilting angle wrt unnormalized weights.
> >
> > --
> >
> > “adversarial vulnerability predates BN. Likewise, non-BN models exhibit adversarial vulnerability. Thus, this title is not a great reflection of the findings of the paper. I would strongly suggest replacing "is a cause" with "increases" or "exacerbates".
> >
> > We are happy to discuss the title of this work. The publication of adversarial examples predating that of batch normalization does not imply that batch normalization cannot act as one of many contributory causes of adversarial vulnerability. Qualitatively, we demonstrated that for MNIST, where a reasonably robust model can be obtained via per-image normalization and L2 regularization (see the adversarial examples of Appendix I which support this), the addition of BN in this case is sufficient to induce adversarial vulnerability. In terms of an apparent relationship between input dimension and vulnerability, which intuitively should not affect robustness as it does not affect the signal-to-noise ratio (Shafahi et al., ICLR 2019), the introduction of batch norm is again sufficient for this adverse relationship to exist. Ultimately, given that robustness for natural datasets is usually a trade-off, e.g., with accuracy (Tsipras et al., 2019), or between minimum and mean margin (Wu & Yu, 2019), we can accept your suggestion to replace "is a cause" with "increases" or "exacerbates".
> >
> > References
> >
> > Dimitris Tsipras and Shibani Santurkar and Logan Engstrom and Alexander Turner and Aleksander Madry. Robustness May Be at Odds with Accuracy. In International Conference on Learning Representations, 2019.
> >
> > Kaiwen Wu and Yaoliang Yu. Understanding Adversarial Robustness: The Trade-off between Minimum and Average Margin. In NeurIPS 19 Workshop on Machine Learning with Guarantees.

---

### Official Review · AnonReviewer2 · 2019-10-22
**Official Blind Review #2**

**Rating:** 3

**Review:**

Overview:
This is an interesting work. The paper is dedicated to studying the effect of BN to network robustness. The author shows that BN can reduce network robustness to small adversarial input perturbations and common corruptions by double-digit percentages. Then, they use a linear "toy model" to explain the mechanism that the actual cause is the tilting of the decision boundary. Moreover, the author conducts extensive experiments on popular datasets to show the robustness margin with or without the BN module. Finally, the author finds that substituting weight decay for BN is good enough to nullify a relationship between adversarial vulnerability and the input resolution.

Strength Bullets:
1. I like the linear toy example. For that binary classification example, the author explicitly explains the boundary tilting, which increases the adversarial vulnerability of the model. It is clear.
2. The paper conducts extensive experiment on SVHN, MNIST, CIFAR10 (C) datasets. And they show performance margin with or without the BN module. And for the attacker setting, they do use the popular setting (i.e. Mardy's PGD setting) in this field which makes the results more convincing.

Weakness Bullets:
1. Why do not visualize the decision boundary of networks (used in this work) to valid the boundary tilting "theory". The toy example is clear but not convincing enough. There exist several techniques may be helpful to the visualization. (i.e. Robustness via curvature regularization, and vice versa). I think it is one of the important parts of this work. The observation of BN causes adversarial vulnerability is interesting but the main focus should be offering more convincing explanations.
2. I do run experiments for VGG11,13,16,19 on cifar10 with PGD 3 attack (Mardy's setting). There exist ~20 ATA performance gaps between networks with BN and BN networks without BN. But for adversarial trained models, the gaps don't exist anymore, at least for VGG11,13,16,19 on cifar10 with PGD 3 attack. (The performance gap is less than 0.5). In other words, without the BN layers, the robustness of adversarially trained models will not increase in my experiments. I see you report some results in Appendix C. But it is not enough to convince me. Could you provide more implementation details about the adversarial training and attacker? And more experiment results about this point are needed. If adversarial training can fix the vulnerability by BN and BN can give a TA boost, there is no reason we need to remove BN in our adversarial training setting. I see there are similar concerns in the OpenReview.
3. [Minior] The experiments need to be organized better. Especially for section 3, it will be better to divide different experiments or observations into the different subsections.


Recommendations:
For the above weakness bullets, this is a week reject.

Suggestions:
1. To solve the weakness bullets;
2. minor suggestion: add the reference mention in the OpenReview, they are related to this work.

Questions:
1. You mention that you run PGD for 20-40 iterations in the experiment at the bottom of page three. But at each table, you only report one number. So my question is for that accuracy number, you run how many iterations for PGD?

**Experience Assessment:**

I have read many papers in this area.

**Review Assessment: Checking Correctness Of Derivations And Theory:**

I carefully checked the derivations and theory.

**Review Assessment: Checking Correctness Of Experiments:**

I carefully checked the experiments.

**Review Assessment: Thoroughness In Paper Reading:**

I read the paper at least twice and used my best judgement in assessing the paper.

---

> ### Author Response · Authors · 2019-11-13
> **Response (part 1) - on visualizations and PGD training**
>
> Thank you for your feedback, and for taking the time to reproduce the main result on several architectures, this effort goes above and beyond, we appreciate it. We respond to each concern below.
>
> --
>
> "Why do not visualize the decision boundary of networks (used in this work) to valid the boundary tilting "theory". The toy example is clear but not convincing enough. There exist several techniques may be helpful to the visualization. (i.e. Robustness via curvature regularization, and vice versa). I think it is one of the important parts of this work."
>
> Aside from space limitations, we wanted to restrict visualizations to scenarios where they are faithful to the underlying model. Although such visualizations would be nice to have for further intuition, we believe the experiments provide sufficient evidence to support the main hypothesis. Also, beyond the toy model, we computed the boundary tilting angle of linear models w.r.t. the nearest centroid classifier on MNIST. Any dimensionality reduction technique used to visualize a high dimensional decision boundary introduces trade-offs that we didn’t feel were necessary to support the main message. We may opt to add such visualizations in a subsequent blog article as you suggest.
>
> --
>
> "The observation of BN causes adversarial vulnerability is interesting, but the main focus should be offering more convincing explanations."
>
> Thank you for stating that you believe the main finding of this work is interesting. We believe we have provided a reasonably accessible explanation, but wish to emphasize that our main contribution is that, to the best of our knowledge, this is the first work to explicitly link batch norm and adversarial vulnerability. This has major implications for state-of-the-art networks, and we believe this connection was previously unknown in the adversarial examples / robustness to distribution shift literature. Many laws of physics that we now take for granted were initially discovered via systematic experiment, and subsequently formalized by others. Other works, e.g., Yang et al., ICLR 2019, Labatie, ICML 2019 examine batch norm at length from a theoretical perspective which supports our conclusions, and a concurrent submission provides further insight as to why the vulnerability we discovered occurs. The official reviews of Yang et al., (https://openreview.net/forum?id=SyMDXnCcF7) expressed concern that their approach was not particularly intuitive or reflecting of popular practice, hence we made an effort to cast the limitations of BN in a practical light for practitioners and researchers of broad backgrounds to understand.
>
> --
>
> "I do run experiments … There exist ~20 ATA performance gaps between networks with BN and networks without BN. But for adversarial trained models, the gaps don't exist anymore, at least for VGG11,13,16,19 on cifar10 with PGD 3 attack. (The performance gap is less than 0.5). In other words, without the BN layers, the robustness of adversarially trained models will not increase in my experiments. I see you report some results in Appendix C. But it is not enough to convince me. If adversarial training can fix the vulnerability by BN and BN can give a TA boost, there is no reason we need to remove BN in our adversarial training setting."
>
> We agree that the adversarial training results originally presented in Appendix C were perhaps a bit too informal. We have tightened up this section, including a full breakdown showing test accuracy and variance for each corruption, instead of simply stating the mean test accuracy over all corruptions.
>
> As we motivated in the main text, it is imperative to consider robustness to unseen adversaries. Thus, it is unfair to benchmark the robustness of natural and adversarially trained networks using the same procedure, when one approach directly optimizes performance w.r.t. one of the evaluations. As you found, in some circumstances the performance degradation of BN seems small if we train on PGD and evaluate on the same, but this no longer holds if we consider other more realistic threat models and common corruptions.
>
> To be more clear, we have renamed Appendix C to “PGD Training Yields Unfortunate Robustness Trade-offs”, as PGD can fail to yield a model with semantically meaningful gradients or convincingly broad robustness even for MNIST, due to the thresholding operation that is learned to favor $\ell_{\infty}$ robustness (see the discussion of Appendix E “MNIST Inspection” of Madry et al., (ICLR 2018)). These limitations have been widely discussed, e.g., Sharma & Chen., (ICLR Workshop 2018), Schott et al., (ICLR 2019), Jacobsen et al., (ICLR 2019, ICLR Workshop 2019), Mu & Gilmer., (ICML Workshop 2019) and are not too surprising given Goodhart’s law: “When a measure becomes a target, it ceases to be a good measure”. The “measure” in this case being the $\ell_{\infty}$ norm of the perturbations, which has become, somewhat arbitrarily, a focal point in the adversarial examples literature.

---

> > ### Author Response · Authors · 2019-11-13
> > **Response (part 2) - Implementation details and references**
> >
> > "Could you provide more implementation details about the adversarial training and attacker?"
> >
> > The meta-parameters used for the PGD training of the WideResNet 28-10 on CIFAR-10-C were  $\epsilon_{\max} = 4/255$, 5 iterations, step size of 1. This was the model for which accuracy on the contrast CIFAR-10-C corruption is degraded by over 20% absolute for Fixup and BN variants. These details were also in a Figure caption, but we have made the connection more explicit in text. We also PGD train normal ResNets {32, 110} using $\epsilon_{\max}=8/255$, 7 iterations, and a step size of 2 (as in Madry et al., 18) and observe similar trends.
> >
> > For PGD training on MNIST we use 20 iterations with a step size $\epsilon / 10$.
> >
> > --
> >
> > Re: Question about 20 or 40 iterations of PGD for eval
> >
> > We agree this wasn’t clear from our language. We report 20 iterations in most cases, but confirmed that 40 iterations did not significantly improve upon this, i.e., degrade the accuracy much further to within the measurement random error. For example, for VGG16 on CIFAR-10 evaluated using 40 iterations of PGD with a step size of $\epsilon_\infty / 20$, instead of 20 iterations with $\epsilon_\infty / 10$, reduced accuracy from $28.9 \pm 0.2\%$ to $28.5 \pm 0.3\%$, a difference of $0.4 \pm 0.5\%$.
> >
> > Also, to err on the side of caution, we use 40 iterations of PGD for evaluation on CIFAR-10 in the new Appendix "On The Initial Learning Rate".
> >
> > --
> >
> > Lastly, as suggested we have discussed the work of Labatie 2019 in several places where relevant.
> >
> > References
> >
> > Yash Sharma and Pin-Yu Chen. Attacking the Madry Defense Model with $L_1$-based Adversarial Examples. In ICLR Workshop, 2018.
> >
> > Lukas Schott, Jonas Rauber, Matthias Bethge, and Wieland Brendel. Towards the first adversarially robust neural network model on mnist. International Conference for Learning Representations, 2019.

---

### Public Comment · ~Anthony_Wittmer1 · 2019-09-28
**Nice work and some questions**

Hi, it is an interesting work.

1. In intuition, the mean and the variance in the BN layer are strongly related with the clean training data, which leads to the unsuitable normalization to the adversarial testing data. I see two related submission, namely https://openreview.net/forum?id=HyxJhCEFDS and https://openreview.net/forum?id=BJlEEaEFDS&noteId=BylDR1y3Pr , where the authors propose MBN to disentangle the distribution for clean and adversarial data to estimate normalization statistics.

2. Adversarially trained models also have the adversarial vulnerability for strong attack. So I have a little doubt whether batch normalization is a cause of adversarial vulnerability for adversarially trained models. In other words, without the BN layers, the robustness of adversarially trained models will increase? In my opinion, the answer is "no", because a stronger neural network architecture can help to increase the robustness of the adversarially trained model.

3. Have the authors tried other normalization layers, which do not need to store the mean and the variance of training data, such as instance normalization, layer normalization and so on?

---

> ### Author Response · Authors · 2019-09-30
> **Thank you for the comments**
>
> 1. Thanks for pointing us to these relevant concurrent submissions. More discussion follows.
>
> 2. Regarding "Adversarially trained models also have the adversarial vulnerability for strong attack", I'm sorry but I don't quite understand the connection to our claim about BatchNorm (BN). All models are vulnerable to strong attacks if "strong" means large perturbation budget. Our claim is simply that using BN makes models more vulnerable. We report results for PGD trained models on MNIST and CIFAR-10 in Appendix C. We take a broad view of robustness, but considering the limited "max-norm" attack model we observe a non-trivial improvement in robustness using Fixup in place of BN for the same WideResNet as in Madry et al. One of the concurrent submissions you reference that aims to fix the vulnerability induced by BN reports an even larger 14% gap in terms of PGD test accuracy for PGD trained vanilla ResNets.
>
> As an aside, I stress that the current form of PGD training with a one-size-fits all epsilon (i.e. the same constant is applied to all examples regardless of their intrinsic noise) can yield excessive invariance that reduces robustness more broadly, see e.g., Jacobsen et al., (2019), Mu & Gilmer, (2019). In the case where epsilon is adapted to each training example, the procedure is equivalent to penalizing the parameters' dual norm for linear models (Xu et al., (2009)). We perform several experiments on the interaction of BN with parameter norm regularization in the text.
>
> 3. We believe it would be valuable to study the effect of these other normalization layers on robustness and would be excited if someone else does so. Due to widespread use of BN and space constraints, these are out of scope for the present submission.
>
> We tested the hypothesis that the vulnerability of BN could arise from using the tracked mean and variance of the training data at test time. This is an insightful observation, indeed not using these statistics at test time does increases robustness, but the same can be achieved through increasing the numerical stability constant (while preserving the ability to test with arbitrary batch sizes). Neither of these mechanisms fully account for the vulnerability, due to decision boundary tilting inherent in the normalization procedure itself. We've added an Appendix to discuss this alternate hypothesis which will become available when updates to all papers are enabled.

---

### Public Comment · ~Antoine_Labatie1 · 2019-10-01
**Two comments**

Hi,

I enjoyed reading your paper. I just have two comments.

First, the discussion and the experiments of Section 3 are very interesting. Do you think that the following interpretation would be valid:
- Since unnormalized MNIST images "live" in a subspace of small dimensionality, many directions do not play any role in the max-margin solution
- On the other hand, normalized MNIST images "live" in a subspace of larger dimensionality, so that many more dimensions play a role in the max-margin solution

Second, I wanted to mention a closely related paper from ICML 2019 precisely showing that batch normalization causes exploding sensitivity to input perturbations at initialization [1].

[1] Characterizing Well-Behaved vs. Pathological Deep Neural Networks. ICML 2019.

---

> ### Author Response · Authors · 2019-10-02
> **Thank you for pointing us to this work**
>
> Yes, your interpretation seems valid, although we do not exactly make this argument in the text. This is worth formalizing and could make our study that considers the input dimension and vulnerability more precise for the case with BN.
>
> We're sorry to have missed the reference you mention from ICML 2019. On first pass, it is indeed highly relevant to our discussion in Section 3, particularly how BN reduces signal-to-noise in Section 6, where you mention "directions of high signal variance are dampened, while directions of low signal variance are amplified". We're looking forward to discussing theses connections in our next revision.

---

> > ### Public Comment · ~Antoine_Labatie1 · 2019-10-03
> > **Thank you for your response**
> >
> > I also believe that the discussion from the ICML 2019 paper on the effect of BN is very relevant. Thank you very much for your quick response.

---

### Public Comment · ~Anthony_Wittmer1 · 2019-10-09
**Minor question**

What is VGG8?

I did not find any clue about the VGG version with 8 weight layers in the paper of VGG.

---

> ### Author Response · Authors · 2019-10-09
> **Architecture**
>
> Thanks for catching this, VGG8 is indeed a non-standard architecture. To reduce capacity, we remove layers from VGG11 such that there is only one convolution layer between each 2x2 max pooling (indicated 'M').
>
> VGG8:   [64, 'M', 128, 'M', 256, 'M', 512, 'M', 512, 'M']
> VGG11: [64, 'M', 128, 'M', 256, 256, 'M', 512, 512, 'M', 512, 512, 'M']

---

### Public Comment · ~Rui_Wang1 · 2019-10-27
**The number of batchnorm layers applied**

Really enjoyed reading your paper, inspired a lot! One question though.

There are models employed not only one but more layers of batch normalizations, have you considered making some evaluation on the relationship between the number of BNs used and the degradation (if any) seen in the same setting as you have demonstrated in the paper?

---

> ### Author Response · Authors · 2019-11-13
> **Robustness generally decreases with many BN layers; effect is slower in residual networks**
>
> Thank you for the question and kind words.
>
> The experiment depicted in Figure 2 examines the impact of increasing the number of layers (where every layer is batch-normalized) from 10-60 in a vanilla feedforward network. There, we see consistency on train-ability with the upper bound from Yang et al., (2019), however in terms of robustness to perturbations/noise there appears to be a sweet spot around the point where accuracy begins to degrade @ ~20-30 layers.
>
> In a separate experiment (not reported in the paper), we train constant width ($n=384$) ReLU networks of 1-10 layers ($L$) with vanilla GD (LR=0.1) on the 3 vs 7 testbed from Section 3. For BNGD over five random seeds, at $L=1$ we have for Clean, AWGN, PGD test sets respectively, absolute test accuracies of $98.4 \pm 0.1\%$, $69 \pm 1\%$, $15.1 \pm 0.8\%$.  For $L=10$, this is reduced to $93.1 \pm 0.2\%$, $52.2 \pm 0.8\%$, and $0.05 \pm 0.01\%$.  Without BN, at $L=1$ we have $98.0 \pm 0.1\%$, $84.7 \pm 0.2\%$, $50.7 \pm 0.7\%$, and for $L=10$, $98.0 \pm 0.1$, $82 \pm 2$, $40 \pm 5\%$.
>
> For BN + SGD (mini batch size 50), the trend is similar but accuracy decays more slowly than for BNGD over the range of 1-10 layers. E.g. for the AWGN test accuracy, instead of dropping from $69 \pm 1\%$ to $52.2 \pm 0.8\%$, dropping from $70 \pm 2\%$ to $66 \pm 4\%$. The biggest factor over this range is between using BN or not, the unnormalized model consistently yields higher AWGN and PGD robustness by a difference of high 10s, and high 20s absolute percent, respectively.
>
> For residual networks, Figure 4 of the paper evaluates the robustness of ResNet{20, 32, 44, 56, 110}, which shows that the robustness gap generally widens with depth between BN and fixed-update initialization (Fixup - Zhang et al., ICLR 2019) variants. Section 7 of Labatie (ICML 2019) sheds theoretical insight as to how the combined action of BN and skip-connections slows down the exploding gradients/sensitivity.

---

### Author Response · Authors · 2019-11-14
**Summary of changes**

Dear reviewers,

Thank you for your thought provoking reviews which have helped us improve the work. Here we briefly recap the main points expressed in the reviews, and where changes have been made to accommodate the concerns raised.

Reviews 2 and 3 are encouraged by the contribution and its significance, but with legitimate concerns about the "gradient masking" phenomenon (R3), and to what extent the results apply to adversarially trained models (R2).

We have demonstrated absence of gradient masking through the following best practices:
- Additive white Gaussian noise (AWGN), and common corruption benchmarks that don't require gradients
- Accuracy versus perturbation magnitude curves (Fig. 5, all go to zero accuracy)
- Unbounded white-box attacks all reach 100% success (fooling images, PGD, CWL2)
- Single step attacks perform worse than iterative attacks (wrt the attacker)
- Black-box transferability analysis on ImageNet (Table 12)
- Qualitative evidence, i.e., adversarial examples that contain semantic features

Regarding adversarial training, we have added commentary and more detailed results showing the limitations of the PGD training approach on common corruptions.

R1 was more sceptical, expressing concerns that the training procedure and effective learning rate may have confounded the results. R1 may not have seen some important details regarding the initial learning rate and schedules that were considered originally, but we can see how this could have been easily missed and have reorganized the section in question. Also, based on their feedback and after reading the linked Li et al., 19 work, we made some interesting discoveries while evaluating robustness on CIFAR-10 every ten epochs during training for different learning rate schedules, which shows more precisely at which times the baselines outperform their BN equivalent.

To more easily parse through the changes, we provide a detailed changelog below:

- Re-organize Section 4 "Empirical Results" (per R1, R3), defer implementation details to Appendix.

- Section 5, clarify relevance of input dimension experiments (per R3)

- Discussed the work of Labatie 2019 where relevant in the Introduction and Section 3.

- Appendix B "PGD Implementation Details" centralize and clarify all implementation details related to PGD and preprocessing (per R2).

- Appendix D "PGD Training Yields Unfortunate Robustness Trade-Offs". Supplementary explanations and full breakdown of MNIST-C results with multiple runs (Table 9), showing that a reasonable baseline with similar clean test accuracy outperforms: BN, PGD, and PGD + BN.

- Appendix I "Alternative Explanations of the Vulnerability". We place the results of a concurrent submission in the context of our work and discussion of BN's numerical stability constant. Qualitatively and quantitatively shows what happens when we "fold" the BN statistics into the weights post-training.

- Appendix J "Adversarial Examples". We craft fooling images and Carlini & Wagner L2 examples on MNIST and SVHN, providing qualitative and quantitative evidence that BN degrades robustness.

- Appendix K "On The Initial Learning Rate". We place our results in the context of Li et al., showing that *at no time during training* does the *best robustness of BN* exceed that obtainable by an equivalent unnormalized model. The BN model's PGD (40-step) accuracy is consistently below 40%, whereas the baseline achieves 52%. We support these results with two standard tests of gradient masking by comparison with AWGN, and FGSM (per R3 concern). Perhaps most interesting is that the PGD accuracy of the BN models declines during training, whereas that of the baseline increases monotonically or plateaus. The use of early stopping is therefore much more important when BN is used, but this still does not recover the robustness of the unnormalized model.

We sincerely hope to have the opportunity to engage in discussions with the reviewers prior to the platform closing in case anything remains unclear, and we thank them for their effort.

---

### Decision · Program_Chairs · 2019-12-19

**Decision:**

Reject

**Comment:**

This article studies the effects of BN on robustness. The article presents a series of experiments on various datasets with noise, PGD adversarial attacks, and various corruption benchmarks, that show a drop in robustness when using BN. It is suggested that a main cause of vulnerability is the tiling angle of the decision boundary, which is illustrated in a toy example.
The reviewers found the contribution interesting and that the effect will impact many DNNs. However, they the did not find the arguments for the tiling explanation convincing enough, and suggested more theory and experimental illustration of this explanation would be important. In the rebuttal the authors maintain that the main contribution is to link BN and adversarial vulnerability and consider their explanation reasonable. In the initial discussion the reviewers also mentioned that the experiments were not convincing enough and that the phenomenon could be an effect of gradient masking, and that more experiments with other attack strategies would be important to clarify this. In response, the revision included various experiments, including some with various initial learning schedules. The revision clarified some of these issues. However, the reviewers still found that the reason behind the effect requires more explanations. In summary, this article makes an important observation that is already generating a vivid discussion and will likely have an impact, but the reviewers were not convinced by the explanations provided for these observations.